# Mitotic deacetylase complex (MiDAC) recognizes the HIV-1 core promoter to control activated viral gene expression

**Emmanuelle Wilhelm**[1], **Mikaël Poirier**[2], **Morgane Da Rocha**[3], **Mikaël Bédard**[4], **Patrick P. McDonald**[5], **Pierre Lavigne**[6], **Christie L. Hunter**[7], **Brendan Bell**[3]*

**1** OSE Immunotherapeutics, Nantes, France, **2** Charles River Laboratories, Sherbrooke, Québec, Canada, **3** Département de microbiologie et d'infectiologie, Faculté de médecine et sciences de la santé, Université de Sherbrooke, and Centre de recherche du CHUS, Sherbrooke, Québec, Canada, **4** Département de Biochimie et de Génomique Fonctionnelle, Faculté de médecine et sciences de la santé, Université de Sherbrooke, and Centre de recherche du CHUS, Sherbrooke, Québec, Canada, **5** Pulmonary Division, Medicine Faculty, Université de Sherbrooke; and Centre de recherche du CHUS, Sherbrooke, Québec, Canada, **6** Département de Biochimie et de Génomique Fonctionnelle, Faculté de médecine et sciences de la santé, Université de Sherbrooke, and Centre de recherche du CHUS, Sherbrooke, Québec, Canada, **7** SCIEX, Redwood City, California, United States of America

* Brendan.Bell@USherbrooke.ca

**Data Availability Statement:** All relevant data are within the manuscript and its Supporting Information files.

## Abstract

The human immunodeficiency virus (HIV) integrates into the host genome forming latent cellular reservoirs that are an obstacle for cure or remission strategies. Viral transcription is the first step in the control of latency and depends upon the hijacking of the host cell RNA polymerase II (Pol II) machinery by the 5' HIV LTR. Consequently, "block and lock" or "shock and kill" strategies for an HIV cure depend upon a full understanding of HIV transcriptional control. The HIV trans-activating protein, Tat, controls HIV latency as part of a positive feed-forward loop that strongly activates HIV transcription. The recognition of the **T**ATA box and **a**djacent **s**equences of **H**IV **e**ssential for **T**at trans-activation (TASHET) of the core promoter by host cell **p**re-**i**nitiation **c**omplexes of **H**IV (PICH) has been shown to be necessary for Tat trans-activation, yet the protein composition of PICH has remained obscure. Here, DNA-affinity chromatography was employed to identify the mitotic deacetylase complex (MiDAC) as selectively recognizing TASHET. Using biophysical techniques, we show that the MiDAC subunit DNTTIP1 binds directly to TASHET, in part via its CTGC DNA motifs. Using co-immunoprecipitation assays, we show that DNTTIP1 interacts with MiDAC subunits MIDEAS and HDAC1/2. The Tat-interacting protein, NAT10, is also present in HIV-bound MiDAC. Gene silencing revealed a functional role for DNTTIP1, MIDEAS, and NAT10 in HIV expression in cellulo. Furthermore, point mutations in TASHET that prevent DNTTIP1 binding block the reactivation of HIV by latency reversing agents (LRA) that act via the P-TEFb/7SK axis. Our data reveal a key role for MiDAC subunits DNTTIP1, MIDEAS, as well as NAT10, in Tat-activated HIV transcription and latency. DNTTIP1, MIDEAS and NAT10 emerge as cell cycle-regulated host cell transcription factors that can control activated HIV gene expression, and as new drug targets for HIV cure strategies.

**Funding:** This work was supported by a CIHR Operating Grant (Funding Reference Number CHR-126637 to BB), and by The Canadian HIV Cure Enterprise Team Grant with funds from the CIHR in partnership with CANFAR and IAS (Grant number HIG-133050 to BB). The FQRNT awarded a research reintegration scholarship to EW. The Alice-E.-Wilson award of the CFUW was awarded to EW. NSERC awarded an Alexander Graham Bell Canada Graduate Scholarship to EW. The NSERC-funded CREATE program, RNA Innovation, awarded a graduate scholarship to MDR. An anonymous donor generously provided financial support to BB. BB is a member of the FRSQ-funded Centre de recherche du CHUS. The funders had no role in study design, data collection and analysis, decision to publish, or preparation of the manuscript.

**Competing interests:** PPMcD is also employed as Executive Director, Research at Insmed Inc. Insmed was however not involved in this work; did not fund it; and does not endorse it, implicitly or otherwise. B.B. owns shares in the biotechnology company Ascioma. Ascioma had no involvement with this research, financial or otherwise.

## Author summary

Latent HIV integrated within the host cell genome poses a major problem for viral eradication. The reactivation of latent HIV depends on host cell transcription factors that are hijacked by the 5' long terminal repeat (LTR) region of the HIV genome to produce viral RNA. At the heart of the LTR of HIV lies a particularly crucial DNA region named the core promoter that is specifically required for the reactivation of HIV by a viral protein named Tat. A significant body of work over more than 30 years has established the specific requirement for the HIV core promoter in Tat's control of HIV latency, but the underlying molecular mechanisms have remained elusive. Here, we identify host cell transcription factors that bind selectively to the HIV core promoter to control activated viral gene expression. Our data reveal three human proteins that act in a complex to reactivate latent HIV, including one that directly recognizes the HIV core promoter, one that regulates chromatin, and a third that binds to the HIV Tat protein. Our data fill a significant and long-standing gap in the understanding of latency and identify new potential drug targets for HIV cure strategies.

## Introduction

The capacity of HIV to rapidly establish latent reservoirs poses a major problem for HIV cure and remission strategies [1]. Additionally, latency is a key obstacle to overcome in the development of an effective therapeutic [2] or protective HIV vaccine [3–5]. Resting memory CD4+ T cells represent the best characterized reservoir of cells harboring latent HIV proviruses [6], but evidence for other cell types as reservoirs has been reported including myeloid cells, microglia, and dendritic cells [7]. A pivotal step in the control of viral latency is viral transcription directed by the HIV 5' LTR of the viral genome (Fig 1A). The HIV LTR orchestrates the expression of viral RNA by recruiting multiple promoter specific transcription factors such as NF-κB [8] and SP1 [9], in concert with host cell chromatin modifying complexes [10,11], ultimately culminating in the formation of the pre-initiation complex (PIC) [12] that contains general RNA polymerase II (Pol II) factors to drive viral transcription [13,14].

Once the first viral RNA is transcribed, the HIV *trans*-activating protein Tat is produced and binds to a stem-loop structure termed TAR at the 5' end of the nascent viral RNA in conjunction with the positive elongation factor P-TEFb (composed of Cyclin T1 and CDK9). Tat recruits P-TEFb from an inactive 7SK small nuclear ribonucleoprotein (snRNP) complex to form an active super elongation complex (SEC), resulting in enhanced Pol II C-terminal domain (CTD) phosphorylation and activation of transcription elongation [15]. The positive feedback loop formed by the action of Tat on the HIV promoter plays a central role in the rapid reactivation of latent HIV [16]. The Tat dependent circuitry that controls latency most probably evolved to permit viral persistence in the face of host immune surveillance [13] and/or enhance early viral transmission during infection [17].

The HIV core promoter, including the TATA box (more precisely the "CATA" box in the case of HIV [18]; Fig 1A), lies at the heart of transcriptional control of HIV latency. Indeed, signals that activate or repress HIV transcription, including extracellular signals, epigenetic events, transcriptional interference, and HIV Tat must converge on the core promoter to impact the formation of the PIC required for transcription initiation [12]. Extensive early mutational studies of the HIV LTR defined the TATA box and adjacent sequences of HIV essential for Tat *trans*-activation (TASHET) of the core promoter (Fig 1A) [19–23]. The

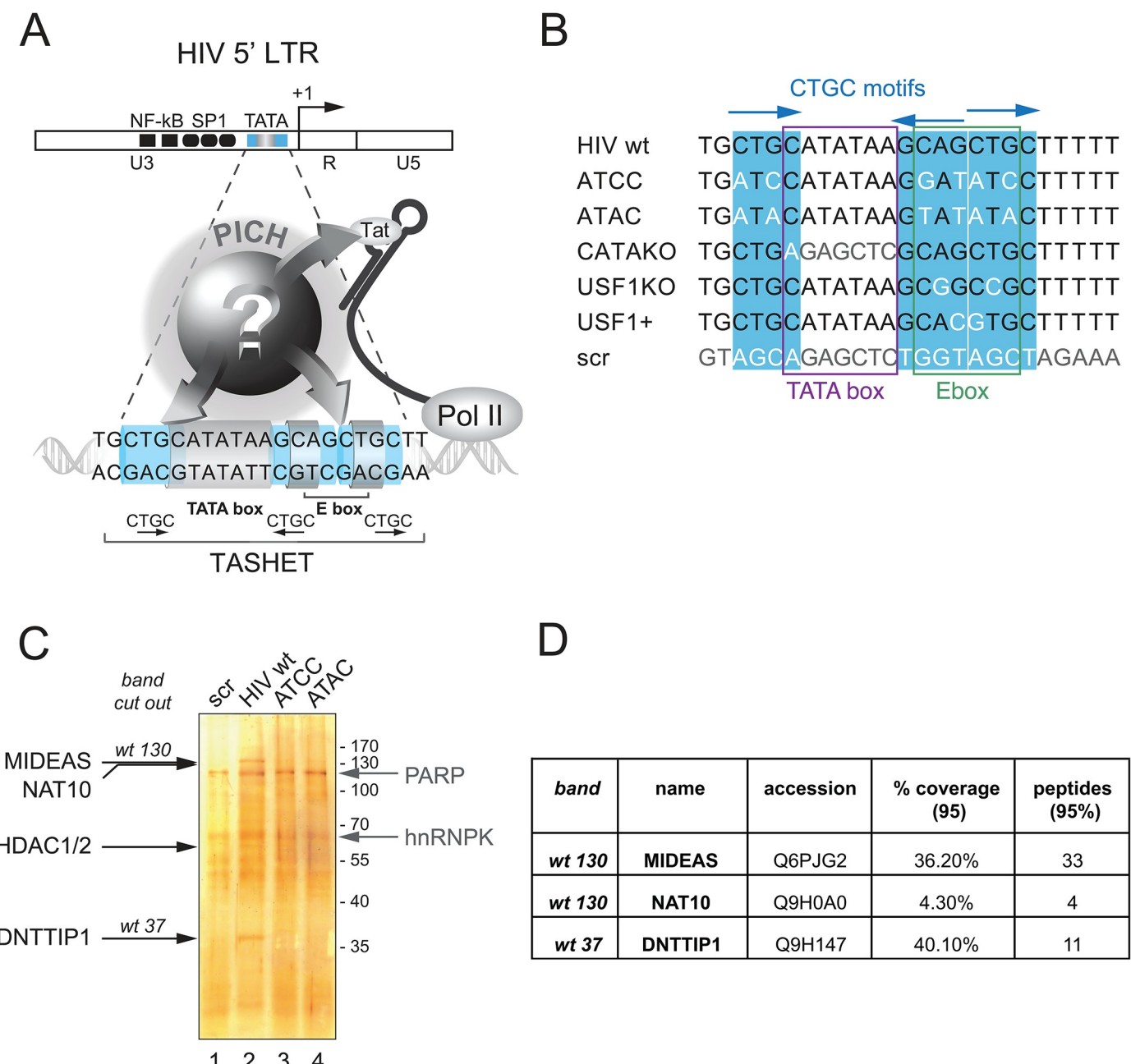

**Fig 1. Affinity chromatography coupled with mass spectrometry to identify TASHET bound factors.** A) The unknown composition of pre-initiation complexes of HIV (PICH). The HIV LTR with transcription factor binding sites is shown at the top with a close-up of the TATA box and adjacent sequences of HIV essential for Tat *trans*-activation (TASHET) shown below. The CTGC DNA motifs (blue) have been shown to be essential for Tat activation and recognition by PICH [25]. PICH contain general transcription factors such as TBP, but their complete composition and how they link TASHET to Tat function have remained unknown (question mark). B) TASHET sequence and point mutations used in this study. The HIV-1 TATA box and E-box are shown by the purple and green squares, respectively. CTGC motifs are highlighted in blue and mutated nucleotides figure in white or light grey. The 'ATCC' mutation was formerly named CTGC5'3' [25]. "Scr" indicates a scrambled control sequence. C) A representative silver-stained polyacrylamide gel reveals the bands corresponding to the proteins specifically (annotated in black) and non-specifically (annotated in light grey) bound to wild type and mutated TASHET. wt 130 and wt 37 stand for the bands that were cut out for mass spectrometry analysis. D) Identity and mass spectrometry scores of the main candidates identified in the analysed bands (see also S2 Fig and S1 Table for further detailed information).

underlying mechanisms conferring Tat responsive transcription selectively upon the HIV core promoter have remained unknown, in part due to the large size, subunit complexity, and dynamic promoter interactions of PIC [24]. More recently, fine mutational analysis of the core promoter revealed that conserved CTGC DNA motifs within TASHET are essential for Tat *trans*-activation in HeLa cells and, more importantly, in primary peripheral blood mononuclear cells (PBMC) [25]. The specific protein composition of PICH that bind to TASHET DNA to allow Tat-mediated HIV transcription is currently unknown.

An in-depth understanding of the mechanisms of HIV transcription is of high strategic value in the search for a cure for HIV/AIDS. Currently, three approaches are being actively pursued towards achieving a cure or remission of HIV infection. These include gene therapy approaches to excise proviral genomes or manipulate HIV transcription using CRISPR/Cas9 based technologies [26]; pharmacological approaches like the "shock and kill" approach [27]; and the "lock and block" strategy [27]. Of these, the shock and kill and lock and block strategies likely hold the most immediate promise to move into clinical application in a timely, safe, and cost-effective manner to patients throughout the world. The shock and kill strategy involves reactivating latent HIV so that it can be recognized and cleared by the host immune system, possibly in concert with immunotherapy [28]. The lock and block strategy involves using small molecules to induce a long-lived state of deep latency that would ideally allow individuals to live without anti-retroviral therapy [29]. The lock and block method has shown promising results in cultured cells [30] and patient lymphocytes *ex vivo* [31], as well as in a humanized mouse model [32], when HIV Tat activity was inhibited using the small molecule cortistatin A. The shock and kill strategy has been employed in numerous clinical trials, although a clinically significant reduction in latent reservoirs has not been achieved [33].

Since viral transcription is a pivotal and rate-limiting step in the exit from or entry into latency, the future clinical success of either the shock and kill or lock and block strategies depends crucially on the ability to selectively control HIV transcription with small molecules. Given that TASHET is a highly conserved viral DNA sequence essential for Tat-mediated HIV transcription [12,25], and that the proof of concept that TASHET can be pharmacologically targeted has previously been established [34], the current gap in our knowledge of PICH composition represents a key question to be answered to pave the way for optimal cure/remission strategies. Here, we have used TASHET DNA affinity chromatography to identify DNTTIP1, MIDEAS and NAT10 as TASHET-binding proteins. We further show that DNTTIP1 and MIDEAS, components of a recently identified mitotic deacetylase complex (MiDAC), play a functionally important role in the recruitment of the Tat-interacting protein NAT10 to impact HIV activated HIV gene expression.

## Results

### Mitotic deacetylase complex (MiDAC) selectively binds to the HIV core promoter

Given the pivotal role that the TASHET element of the HIV core promoter plays in the transcriptional control of HIV latency and the response to Tat *trans*-activation, we set out to identify host cell factors that specifically bind to TASHET DNA (Fig 1A). We employed biotinylated double-stranded synthetic oligonucleotides containing the wild type TASHET sequence or TASHET bearing point mutations (Fig 1B), as controls for DNA binding specificity. Nuclear extracts from HeLa cells were incubated with TASHET DNA immobilized on streptavidin-beads followed by extensive washes. Cellular TASHET-binding proteins were then denatured and fractionated by SDS-PAGE and analyzed by silver staining (Fig 1C). To identify novel TASHET-binding proteins, two major protein bands that migrated with relative

molecular weights of 130 kDa and 37 kDa (Fig 1C, lane 2) were excised, treated with trypsin, and subjected to analysis by liquid chromatography coupled to tandem mass spectrometry (LC-MS/MS). Importantly, these proteins were not found in affinity purifications using two distinct TASHET sequences where the CTGC motifs were mutated to ATCC or ATAC (Fig 1C, lanes 3 & 4), point mutations that have been shown to block Tat *trans*-activation in cells, and to disrupt normal PICH formation upon TASHET *in vitro* [25] (S1 Fig). To identify high confidence TASHET binding proteins, we filtered the peptide data to remove technical artifacts (eg. keratin, trypsin), and proteins such as PARP1 and hnRNPK that bound non-specifically to a scrambled oligonucleotide (Fig 1C, lane 1). After filtering, the peptides from the 130 kDa band corresponded to proteins MIDEAS and NAT10, and for the 37 kDa band, protein DNTTIP1 (TdIF1/C20orf167) (Fig 1D), see also S2 Fig and S1 Table for details of the mass spectrometric identification of peptides. MIDEAS (ELMSAN1/C14orf43) and DNTTIP1 both form part of a histone deacetylase complex (HDAC) termed MiDAC whose formation and activity increase during mitosis [35, 36]. Intriguingly, NAT10 was identified as an acetyltransferase [37], and interacts with the viral *trans*-activating protein Tat *in vitro* [38, 39] and *in cellulo* [39]. These data suggest that the MiDAC subunits MIDEAS and DNTTIP1, as well as the Tat-interacting protein NAT10, selectively bind to the wild type HIV TASHET sequence.

To independently confirm the binding of MIDEAS, DNTTIP1 and NAT10 to TASHET, we used immunoblotting detection with specific antibodies to analyse proteins enriched from HeLa cell nuclear extracts by TASHET affinity chromatography. MIDEAS and DNTTIP1 bound to wild type TASHET (Fig 2A, lane 3) but not to scrambled oligonucleotides (Fig 2A, lane 2) or to mutated TASHET (Fig 2A, lanes 4 & 5). Because the MiDAC complex contains HDAC1 [35, 36], we also tested whether HDAC1 can be recovered by TASHET DNA chromatography. Indeed, HDAC1 was found to interact with wild type TASHET but not with mutated TASHET (Fig 2A, lane 3 versus 4 & 5). As a positive control for the sensitivity and selectivity, we tested for the presence of the transcription factor AP4 that has previously been shown to bind to the E-box of the HIV core promoter [40]. We found that AP4 bound to wild type TASHET but not to TASHET containing mutations that disrupt the E-box (Fig 2A, lane 3 versus lanes 4 & 5; Fig 1B). Overall, our results reveal that MiDAC complex subunits MIDEAS, DNTTIP1 and HDAC1 selectively interact with the wild type TASHET element of the HIV core promoter.

To extend the observations from the HeLa model cell line to CD4+ lymphocytes, we prepared nuclear extracts from the Jurkat T lymphocyte cell line. When T cell nuclear proteins were enriched by TASHET-affinity chromatography, the MiDAC subunits MIDEAS and DNTTIP1 bound to the wild type but not mutated TASHET (Fig 2B, lane 3 and 4, respectively). The result shows that MIDEAS and DNTTIP1 are expressed in human T lymphocyte cell lines, and that they bind selectively to the HIV core promoter TASHET DNA element. To further extend our observations to primary cells, we isolated nuclear extracts from PBMC from uninfected donors. We tested the binding of MiDAC subunits MIDEAS, DNTTIP1 and HDAC1 to TASHET and found that all of them bound selectively to HIV TASHET (Fig 2C, lane 3). We employed nuclear extracts from PBMC to further define the DNA binding specificity of MiDAC components. TASHET mutations that prevent Tat responsiveness were bound poorly by MIDEAS, DNTTIP1 and HDAC1 (Fig 2C, lanes 4 & 5). Mutation of the HIV TATA box reduced binding of MIDEAS, DNTTIP1 and HDAC1 (Fig 2C, lane 6), implying a significant role for the TATA sequence in their binding to TASHET. The CTGC motifs required for MiDAC binding to TASHET *in vitro* and Tat *trans*-activation *in cellulo* overlap with an E-box motif 5' of the TATA box (Fig 1B). To dissect the binding specificity of the MiDAC complex from that of bHLH factors such as USF1 and AP4, we employed point mutations within the E-box that block USF1 binding to TASHET and yet remain responsive to Tat

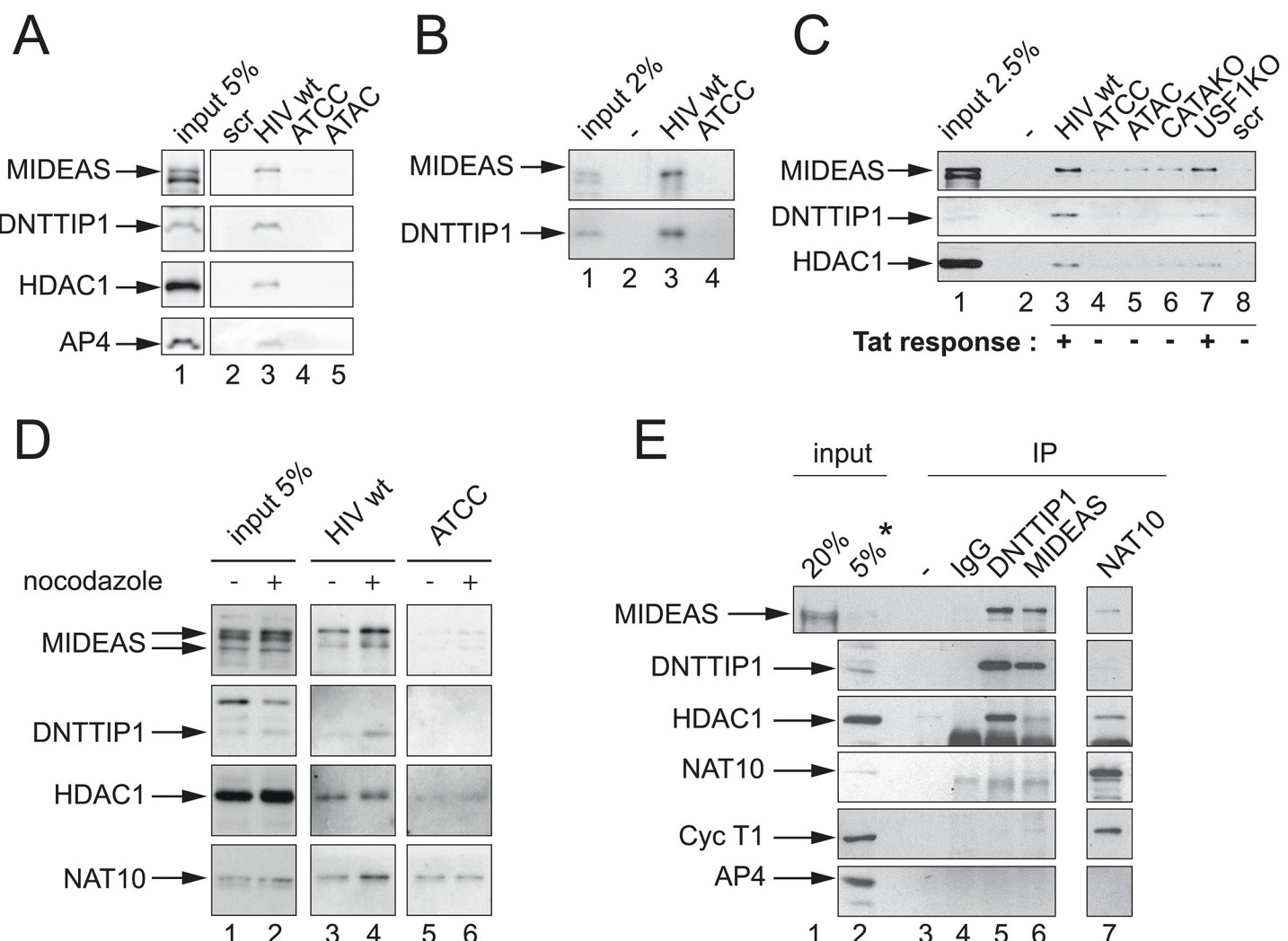

**Fig 2. MIDEAS and DNTTIP1 form a complex that specifically binds to TASHET.** A) Western blot validation of the binding of MIDEAS, DNTTIP1 and HDAC1 from HeLa cell nuclear extracts to wild type TASHET. B) As in A, except with nuclear extracts from Jurkat cells. In the case of NAT10, note that levels in lanes 5 & 6 correspond to background levels that are observed also with random DNA sequences. C) As in A, except with nuclear extracts from PBMC. MiDAC components selectively bind wt TASHET and retain binding for USF1KO mutant. Tat response as assayed in reporter transfected cells is expressed as '+' when > 90% and '-' when <20% response to Tat transactivation [25, 111]. D) As in A, except that the HeLa cells were blocked in the G2/M phase by a 16h treatment with nocodazole before nuclear protein extraction. E) Co-immunoprecipitation showing interactions among MiDAC components, as well as with Cyclin T1 in Jurkat nuclear extracts. Antibodies used for IP are indicated at the top, antibodies used in Western Blot to the left. For visibility, co-IP are compared to 20% (lane 1) or 5% (lane 2) input, except for DNTTIP1, where lane 2 (indicated by *) corresponds to 10% input.

*in cellulo* [25]. One such mutation, USF1 KO (Fig 1B), retained significant amounts of MID-EAS, DNTTIP1 and HDAC1 (Fig 2C, lane 7), consistent with an essential role for MiDAC, but not USF1, in Tat *trans*-activation.

MiDAC can be distinguished from other known HDAC complexes by its atypical property of displaying its highest deacetylase activity when purified from mitotic cell extracts [35]. We therefore tested whether the newly identified TASHET-binding proteins differentially interacted with the HIV promoter during mitosis. HeLa cells were arrested in mitosis by treating with nocodazole, and nuclear extracts were prepared. Following TASHET-affinity chromatography, enriched proteins were analysed by immunoblotting. MIDEAS, DNTTIP1, HDAC1 and NAT10 all bound more efficiently to TASHET when purified from mitotic versus non

arrested cells (Fig 2D, lane 4 versus 3). NAT10 from mitotic cells bound more avidly to the wild type TASHET than to TASHET bearing Tat-unresponsive mutations (Fig 2D, lane 4 versus 6). Nonetheless, NAT10 showed background binding to mutated TASHET (Fig 2D, lanes 5 & 6), likely due to the documented non-specific nucleic acid binding properties of NAT10 [41]. We conclude that the binding of NAT10 to TASHET is sensitive to cell cycle regulation with higher affinity during mitosis. Significantly, these results show that the selective interaction of MIDEAS, DNTTIP1, HDAC1 and NAT10 is enhanced during mitosis.

MIDEAS, DNTTIP1, HDAC1 and NAT10 (particularly from mitotic cells) were found to bind to TASHET, all with indistinguishable binding specificities (Fig 2A, 2C and 2D). Additionally, previous work has demonstrated that MIDEAS, DNTTIP1 and HDAC1 can all be found within the MiDAC complex in the absence of DNA [35, 36, 42]. Together, these findings suggest that the newly identified TASHET-binding proteins may be recruited to TASHET as a preformed complex. To determine whether complexes containing MIDEAS, DNTTIP1, HDAC1 and NAT10 can form in the absence of TASHET DNA, we performed co-immuno-precipitation experiments. We found that DNTTIP1 associated strongly with MIDEAS and HDAC1 (Fig 2E, lane 5). Likewise, MIDEAS co-immunoprecipitated with DNTTIP1 and HDAC1 (Fig 2E, lane 6). NAT10 was found associated with MIDEAS and HDAC1 (Fig 2E, lane 7). Because NAT10 reportedly interacts with HIV Tat [38, 39], we tested the association of the established Tat cofactor, cyclin T1, with TASHET-binding proteins. Cyclin T1 associated with NAT10 (Fig 2E, lane 7) and weakly with MIDEAS (Fig 2E, lane 6). Based on the substoichiometric interaction of the Tat-interacting protein with both MIDEAS and TASHET, we propose that NAT10 is recruited to TASHET indirectly, possibly through MIDEAS, and preferentially during mitosis. Our results show that DNTTIP1 and MIDEAS interact strongly in the absence of DNA and that NAT10 associates with MIDEAS, directly or indirectly. Taken together, the above results reveal a selective binding of MiDAC components MIDEAS, HDAC1 and DNTTIP1, in substoichiometric association with NAT10, to TASHET DNA.

## DNTTIP1 directly recognizes the HIV core promoter *in vitro*

Having established that several MiDAC components bind to the HIV core promoter, we next sought to determine how the complex selectively engages TASHET DNA. Upon inspection of the known domains of HDAC1, MIDEAS, NAT10 and DNTTIP1 using the Conserved Domain Database (CDD) of the National Centre for Biotechnology Information (NCBI) [43], only DNTTIP1 was found to contain domains known to confer sequence specific DNA binding including both an AT-hook domain [44–46] and a domain initially predicted by Kubota and colleagues [47] to contain a helix-turn-helix (HTH) domain whose structure has now been solved and shown to related to the SKI/SNO/DAC domain [42,48,49] (Fig 3A). In fact, Kubota and colleagues have shown that DNTTIP1 binds to an AT-rich motif in conjunction with the specific sequence 5'-G**NTGC**ATG-3' using SELEX *in vitro* [50], and that DNTTIP1 can also recognize similar motifs using ChIP-Seq (ChIP-Sequencing) *in cellulo* [51]. When we aligned the binding motifs defined by SELEX to the TASHET sequence containing the TATA box and flanking CTGC sequences, sequence similarities were observed (Fig 3A), raising the possibility that DNTTIP1 could bind TASHET directly.

To test whether DNTTIP1 can bind to TASHET directly, we purified recombinant human His-tagged DNTTIP1 expressed in bacteria (Fig 3B). After purification, the secondary structure content of DNTTIP1 was subjected to analysis by circular dichroism (CD) in order to ascertain that it is folded. The CD spectrum showed a characteristic minimum at 222 nm (Fig 3C), indicating that there is high α-helical content in the protein, a finding that is compatible with the structures of the N- and the C-terminal domains of DNTTIP1 that have recently been

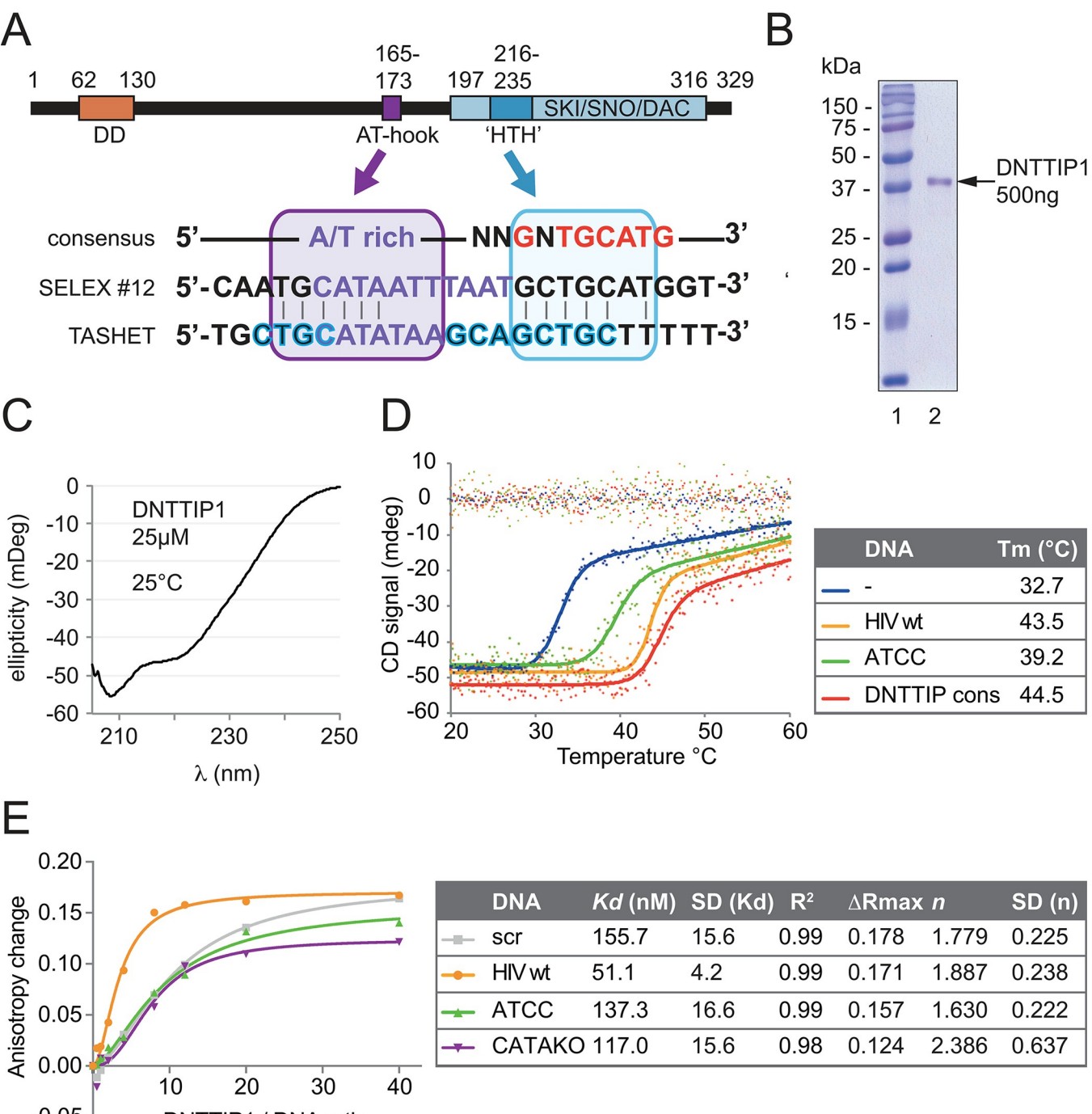

**Fig 3. DNTTIP1 interacts directly, with high affinity and specificity to wt TASHET.** A) Upper panel: schematic view of DNTTIP1 protein showing the dimerization domain (DD in orange), AT-hook (purple) and c-terminal DNA binding domain (in pale blue) structurally related to the SKI/SNO/DAC domain [42]. The SKI/SNO/DAC domain contains an embedded motif proposed to have structural similarity to helix-turn-helix (HTH) motifs [47, 48] (dark blue). Lower panel: alignment of the DNTTIP1 consensus sequence (top), the SELEX sequence #12 as published by Kubota et al. [50], and the TASHET sequence. High similarity sequences are shown including the 5'CTGC and TATA motifs (purple box) and around the 3' CTGC motif (blue box). Vertical lines indicate identities. B) Coomassie staining, and C) Circular Dichroism (CD) spectroscopic spectrum of purified recombinant His-tagged DNTTIP1. D) CD spectroscopy thermal denaturation curves measured at a 222nm wavelength (left) and calculated melting temperatures (Tm) (right) of recombinant DNTTIP1 in absence (blue) or equimolar presence of wt (yellow), or ATCC (green) mutated TASHET, or of a DNTTIP1 binding consensus sequence (sequences in S1 File) [50] (red). D) Anisotropy changes of fluorescein labeled DNA (sequences in Fig 1 and S1 File) in presence of increasing concentration of recombinant DNTTIP1. Measured values have been fitted to a specific binding with Hill slope (left panel: graphic representation), yielding the calculated Kd values with its SD, high $R^2$ (right panel), ΔRmax corresponding to the plateau, n corresponding to the Hill coefficient with its SD.

solved [42]. Indeed, the N-terminal domain was shown to form a tight dimerization domain with a novel and all-α architecture, and the C-terminal domain was observed to adopt a tertiary structure related to the SKI/SNO/DAC DNA binding domain that is also highly α-helical. We note that the raw numerical data throughout this work can be found in S1 Data. To verify that the recombinant protein was stably folded, we recorded its temperature denaturation monitored by CD at 222 nm. The cooperative nature of the denaturation curve obtained clearly indicates the existence of folded α-helical structure stabilized by tertiary and quaternary interaction as expected from the structures reported for the N and C-termini of DNTTIP1 (Fig 3D, in blue). The α-helicity of recombinant DNTTIP1 allowed us to interrogate a potential interaction between DNTTIP1 and TASHET DNA by comparing the thermal denaturation of the protein in the absence and presence of DNA [52,53]. In such experiments, the observation of a shift in the thermal denaturation curve towards higher apparent melting temperatures (Tm) indicates a favorable binding event. In fact, the favorable binding provides stabilisation free energy to the protein that moves the binding curve to a higher Tm. The shift in Tm is directly proportional to the apparent affinity of a protein for a DNA probe [52]. As a positive control, incubation of DNTTIP1 with an equimolar concentration of an oligonucleotide known to bind it with high affinity [50] significantly stabilized the protein when compared to protein alone (Fig 3D, red versus blue). When DNTTIP1 was incubated with TASHET DNA, it was also stabilized shifting from a Tm of 32.7˚C to 43.5˚C (Fig 3D, yellow versus blue). The strong stabilization of DNTTIP1 in the presence of TASHET DNA shows that DNTTIP1 can directly interact with TASHET in solution. To ask whether the binding of DNTTIP1 to TASHET was specific, we incubated DNTTIP1 with an oligonucleotide bearing Tat-unresponsive point mutations within the CTGC motifs (Fig 1B), and observed a significantly more modest increase in the thermal stabilization by this DNA to only 39.2˚C (Fig 3D, yellow versus green). The CD results demonstrate a direct and selective interaction of DNTTIP1 with TASHET that depends on intact CTGC motifs.

To obtain more quantitative estimates of the affinities (apparent dissociation constant—$K_D$) on the DNTTIP1 –TASHET interaction, as well as to reconfirm the interaction by an independent biophysical technique, we employed fluorescence anisotropy to probe the interaction [54]. Titration of fluorescently labeled wild type TASHET with increasing concentrations of DNTTIP1 increased the fluorescence anisotropy, as expected for the formation of a stable complex which reduces the tumbling of the fluorescent probe (Fig 3E, yellow curve). To calculate the apparent $K_D$ of the wild type TASHET–DNTTIP1 interaction, we used the Hill equation [55] to simulate binding curves. More precisely, the change in anisotropy (ΔR) was simulated using the following equation: $\Delta R = \Delta R_{Max} \times ([DNTTIP1]^n / (K_D + ([DNTTIP1]^n))$ to determine $\Delta R_{Max}$, n (the Hill coefficient) and $K_D$ by non-linear least-squares fitting. Under our experimental conditions, DNTTIP1 bound to TASHET with a dissociation constant ($K_D$) of 51.1 nM (Fig 3E). The sigmoidal nature of the change in anisotropy, and the Hill coefficient of 1.89, together indicate a cooperative binding mode and further documents that DNTTIP1 interacts with TASHET in a multimeric form (Fig 3E). In light of the fact that DNTTIP1 has been shown to contain a dimerization domain [42], it is tempting to ascribe a Hill coefficient close to 2 to a dimeric form of DNTTIP1, however it is important to bear in mind that a Hill coefficient larger than 1 can only rigorously be interpreted as an indication of cooperativity between the oligomerization of DNTTIP1 and DNA binding as the concentration of the former increases [55]. To assess the specificity of the DNTTIP1 –TASHET interaction, the CTGC motifs of TASHET were mutated to ATCC, since this mutation prevented MiDAC binding (see Fig 2A above). TASHET bearing the ATCC mutation had a significantly reduced affinity for DNTTIP1 yielding a calculated $K_D$ of 137.3 nM (Fig 3E). Mutation of the TATA box also reduced, albeit to a lesser extent ($K_D = 117.0$ nM), the binding of DNTTIP1. Finally, the lowest

affinity for DNTTIP1 measured was that of a random sequence oligonucleotide ($K_D$ = 155.7 nM). Consistent with published data, this result shows that DNTTIP1 possesses both non-specific [42], and sequence-specific [50, 51] DNA binding activities. Taken together, the biophysical data from the CD and anisotropy experiments demonstrate a direct and specific interaction with TASHET that depends on its flanking CTGC and TATA box DNA sequences.

## DNTTIP1 occupies the HIV promoter *in cellulo*

The direct interaction of DNTTIP1 with TASHET via the CTGC and TATA box sequences *in vitro* is compatible with a model whereby DNTTIP1 is the major DNA binding subunit responsible for recruitment of other MiDAC subunits to the HIV core promoter. We next asked whether DNTTIP1 can interact with the HIV core promoter within an integrated provirus *in cellulo* using chromatin immunoprecipitation (ChIP). We initially tested numerous commercial antibodies against DNTTIP1 in ChIP but were unable to identify antibodies that performed well in ChIP. To circumvent the lack of appropriate antibodies, we used a lentiviral vector to introduce a Flag-tagged DNTTIP1 into HeLa cells that express DNTTIP1-Flag at levels comparable to those of endogenous DNTTIP1 (Fig 4A, lane 4). We then infected these cells with a pseudo-typed HIV reporter virus capable of a single round of infection [25]. ChIPs were then performed to test the presence of DNTTIP1 and known transcription factors upon the HIV promoter using two distinct PCR primer sets targeting the core promoter and a more distal region of the HIV LTR (Fig 4B). In positive control ChIPs, Pol II occupancy was detected upon the core and distal regions of the HIV LTR and an active cellular gene, *DDIT3*, but not upon the gene desert *GDM*, or the inactive *β-globin* promoter (Fig 4C). Likewise, the general transcription factor TBP was also found to associate with the HIV LTR and active cellular *DDIT3*, but not with negative control region *GDM* or the inactive *β-globin* promoter (Fig 4C). HDAC1 was enriched upon the HIV LTR and was present at substantially lower levels on the active promoter of *DDIT3*, as expected (Fig 4C). Likewise, HIV Tat specifically interacted with the HIV LTR but not the active cellular *DDIT3* promoter (Fig 4C). ChIP assays to detect DNTTIP1-Flag revealed its enrichment upon the HIV LTR (core and distal elements), but its absence upon the active cellular *DDIT3* promoter (Fig 4C). The ChIP data therefore demonstrate that the specific interaction of DNTTIP1 with TASHET *in vitro* also occurs with DNTTIP1 and the proviral HIV promoter *in cellulo*.

## DNTTIP1, MIDEAS and NAT10 are required for full HIV promoter activity

The specific binding of MiDAC subunits DNTTIP1, MIDEAS and their substoichiometrically associated Tat interacting protein, NAT10, to TASHET raised the possibility that these nuclear proteins play a functional role in HIV transcription. To test whether these proteins impact HIV expression in living cells, we depleted their expression levels using small interfering RNAs (siRNAs) in HeLa cells. A readily detectable reduction in target protein expression was obtained for at least two independent siRNAs against MIDEAS, DNTTIP1, and NAT10 (Fig 5A). After depleting MIDEAS, DNTTIP1 and NAT10 proteins, we transfected the cells with a reporter plasmid in which the HIV LTR drives eYFP reporter gene expression with or without a Tat expression vector. Basal HIV transcription, in the absence of Tat, was statistically significantly reduced by treatment with two of three siRNAs targeting MIDEAS, two siRNAs targeting DNTTIP1 and two siRNAs targeting NAT10 (Fig 5B). Tat *trans*-activated HIV transcription was significantly reduced by all siRNAs depleting MIDEAS, DNTTIP1 and NAT10 (Fig 5C). We conclude that DNTTIP1, MIDEAS and NAT10 are important for basal and especially for Tat activated HIV transcription in living cells.

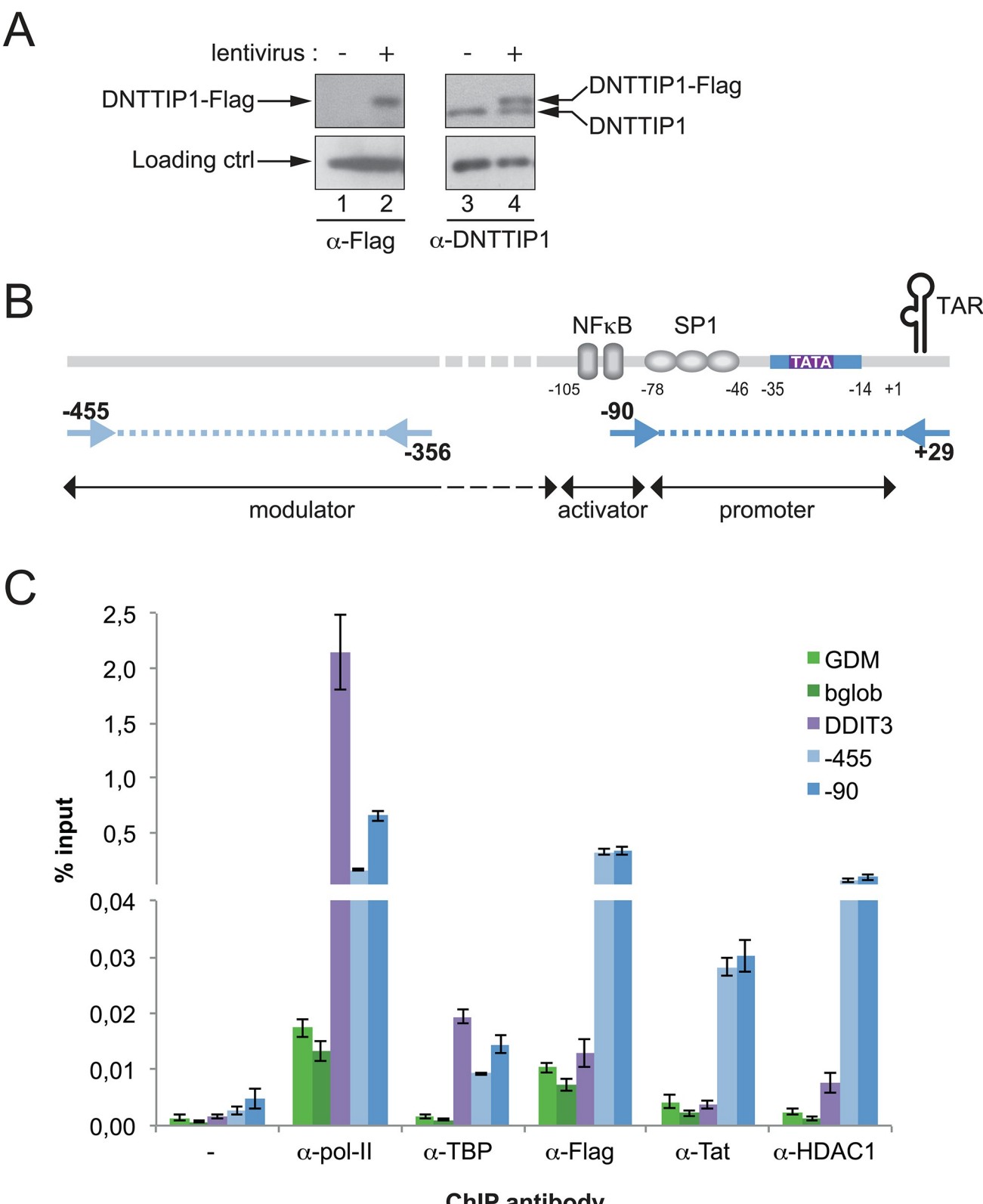

**Fig 4. DNTTIP1 occupies the HIV proviral promoter *in cellulo*.** A) Western Blot showing the expression level of exogenous DNTTIP1-Flag transgene compared to endogenous DNTTIP1 in a cell population with stable integration of a lentivirus bearing the DNTTIP1-Flag sequence (+) vs an empty lentivirus (-). TBP was used as a loading control. B) Positioning of the qPCR primers used in ChIP qPCR on HIV LTR region. C) Chromatin

Immunoprecipitation in NL4.3-Luc infected HeLa cells expressing a stable version of flagged-DNTTIP1. Enrichment obtained with antibodies directed against Pol-II, TBP, Flag-DNTTIP1, HIV1-Tat and HDAC1 vs no antibody (-) on inactive chromatin regions (GDM, bGlob, green bars), an active cellular gene promoter (DDIT3, purple bars), integrated HIV distal (-455, light blue bars) and proximal (-90, blue bars) promoter. The error bars represent the standard deviation of three independent ChIP assays performed in parallel. The chromatin occupancy is expressed in percentage of input.

To validate a role of newly identified TASHET-binding proteins in a cellular context more like the primary physiologic target of HIV, human T helper cells, we employed the CD4+ and HIV susceptible lymphoblast cell line, MOLT4 [56]. We focused on the central DNA binding subunit of MiDAC, DNTTIP1, and used lentiviral transduction to produce shRNA for efficient depletion of DNTTIP1 that was evaluated by immunoblotting and quantitation (Fig 6A and 6B). Cells were subsequently infected with single-round reporter viruses and HIV gene expression was assessed by luciferase expression [25]. The depletion of DNTTIP1 protein in MOLT4 lymphoblasts by three independent shRNAs resulted in a statistically significant decrease in HIV reporter expression (Fig 6C) but had no impact on the number of integrated HIV proviruses (Fig 6C, green). We conclude that the DNTTIP1 subunit of MiDAC, that selectively recognizes TASHET within the HIV core promoter, is required for full HIV transcription in human lymphocytes.

## Activation of HIV gene expression by latency reversing agents (LRA) acting through the P-TEFb/7SK axis require the DNTTIP1 binding sites of TASHET

To determine if the binding of DNTTIP1 to TASHET via the CTGC motifs plays a role in the activation of latent HIV in human lymphocytes, we infected Jurkat T lymphocytes with wild type single-round reporter virus or a virus with mutations that block DNTTIP1 binding and then treated them with known reversing agents (LRAs) (Fig 7A). Previous work has established that in such experimental settings the large majority of infections rapidly enter a transcriptionally inactive state in cell lines, including Jurkat lymphocytes [57, 58], as it also does in both resting and activated primary CD4+ cells [59]. The choice use of this cellular model of HIV gene activation to evaluate the impact of the CTGC motifs for two reasons. Firstly, the chromatin environment into which HIV integrates can profoundly affect HIV expression levels [60]. To avoid potential artifacts due to clonal selection of a specific integration site, we infected populations and measured the aggregate outcome on HIV gene expression. Secondly, unlike established long-term models of HIV latency, this system permits the mutation of nucleotides required for binding to the newly identified regulators of HIV gene expression.

As expected, the expression of the wild type virus was significantly activated in response to all known LRAs including tumor necrosis factor-alpha (TNF-α), prostratin, romidepsin, and HMBA (Fig 7B), showing that the system is responsive to latency reversing agents as previously reported [57, 58]. We then tested the impact of TNF-α and prostratin, that are both known LRAs that act primarily on HIV transcription via activation of NF-κB [61, 62] (Fig 7A). Both treatments significantly increased HIV reporter expression in a manner highly dependent on intact NF-κB binding motifs (Fig 7B), confirming that the viral reporter system is sensitive to transcription factor-specific point mutations. We next tested mutations that abolish the binding of the general transcription factor, TATA binding protein (TBP). As expected, the promoter lacking a functional TATA box was crippled and responded poorly to all LRAs, with only modest inductions in response to the NF-κB-dependent stimuli, TNF-α and prostratin (Fig 7B). The activation of HIV expression by romidepsin, a broad spectrum HDAC inhibitor [35] (Fig 7A) that can activate latent HIV *in vivo* [63] was not significantly changed by mutations that block binding of DNTTIP1 (Fig 7B). In contrast, activation of HIV by HMBA,

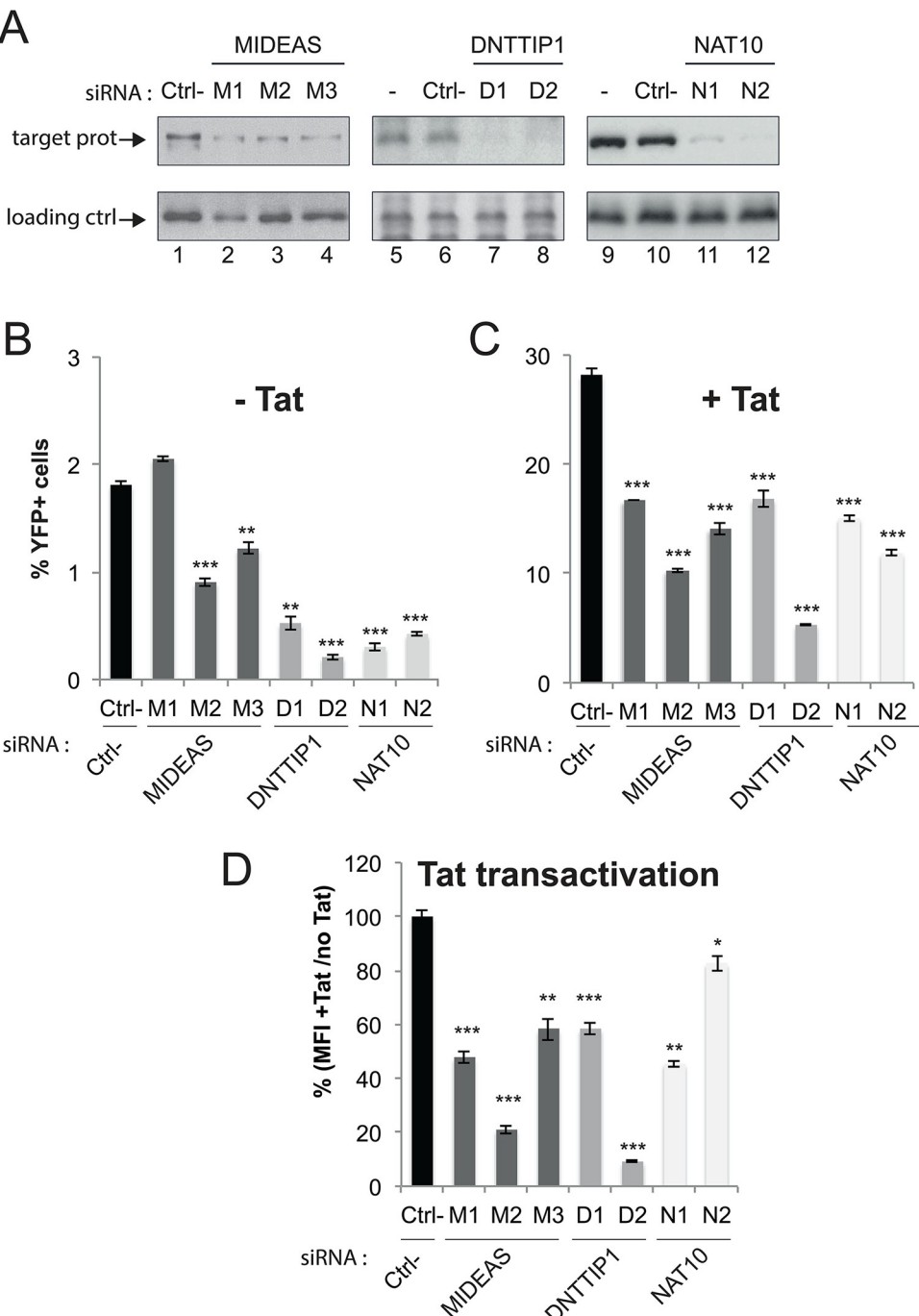

**Fig 5. DNTTIP1 and MIDEAS are required for optimal HIV-1 promoter activity in living cells.** The activity of a transfected HIV-1 promoter reporter was measured in HeLa cells after siRNA mediated knock-down of DNTTIP1 (D1 and D2 siRNAs), MIDEAS (M1, M2, and M3 siRNAs) or NAT10 (N1 and N2 siRNAs) compared to a non-targeting siRNA (ctrl-) or mock transfected cells (-). A) Target Knock-Down validation by Western Blot. USF1 protein levels were monitored as a loading control. (B) The percentage of YFP positive cells in the absence (cells co-transfected empty vector control) of HIV Tat. (C) The percentage of YFP positive cells in the presence of HIV Tat (cells co-transfected Tat expressing plasmid) of HIV Tat. P-values were calculated in comparison to the Ctrl siRNA (black bars). * p<0.05, **p<0.01, ***p<0.001. D) Tat transactivation that is expressed as the ratio of mean fluorescence intensity in presence vs that measured in the absence of Tat co-expression, with siRNA ctrl being set to 100%. The error bars in panels B–D represent the standard deviation of three independent experiments performed in parallel.

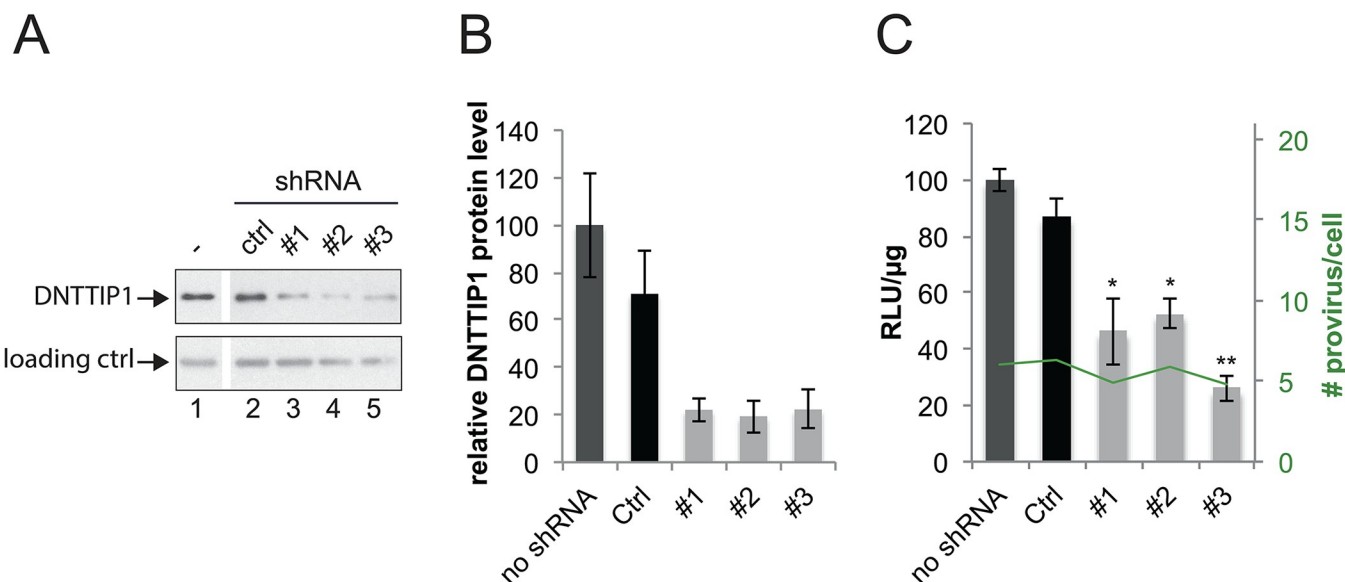

**Fig 6. DNTTIP1 is required for the optimal activity of an integrated provirus in MOLT4 lymphocytes.** The activity of integrated proviral HIV-1 promoter was monitored by luciferase activity after knock-down of DNTTIP1 by three different shRNA lentiviral constructs (#1, #2, #3). A) Target Knock-Down validation by Western Blot. Tubulin was employed a loading control. B) Quantification of relative DNTTIP1 protein expression measured in A. C) The integrated proviral HIV promoter activity (left axis: RLU/μg, the mock control being set to 100%), and the number of HIV proviruses per cell as analyzed by qPCR (right axis, in green) were measured in the same cells. Error bars represent the standard deviation of three independent infections performed in parallel. P-values were calculated in comparison to the Ctrl shRNA (black bars). * p<0.05, **p<0.01.

which acts on the 7SK small ribonucleoprotein complex (7SK snRNP) by releasing active P-TEFb from HEXIM1 [64,65] (Fig 7A) was completely lost when the DNTTIP1-binding motifs were mutated (Fig 7B), indicating that the activation of HIV by least some LRAs depends on DNTTIP1 binding in living cells.

To further dissect the contribution of DNTTIP1 to the activation of HIV gene expression, we employed selective point mutations to the DNTTIP1 binding motifs and a broader spectrum of LRAs. We employed TNF-α as a positive control that enhanced HIV expression in all cases, except when reporters contained mutated NF-κB sites, as expected (Fig 7C). Indeed TNF-α disrupted HIV latency more efficiently for viruses bearing point mutations to the DNTTIP1 binding sites (Fig 7C). Activation by the HDAC inhibitor SAHA was modestly impaired when the DNTTIP1 binding sites were mutated (Fig 7C), indicating that DNTTIP1 binding can contribute to SAHA-mediated HIV activation. The most striking finding was that mutation of the DNTTIP1 binding sites strongly impaired the reactivation of HIV expression by HMBA, JQ1 and MMQO (Fig 7B and 7C). Importantly, HMBA, JQ1 and MMQO all act through the P-TEFb/7SK pathway (Fig 7A). JQ1 antagonizes BRD4 to release active P-TEFb from the 7SK snRNP complex [66, 67]. MMQO was identified in a screen for selective HIV LRAs [68], and like JQ1 acts by displacing BRD4 [69]. To ensure that the impact of the mutations was due to the loss of DNTTIP1 binding and not to the spurious binding of additional factors, we used an independent mutation (ATAC) that produces less binding than the wild type TASHET in EMSA with Jurkat or HeLa cell nuclear extracts (S1A Fig, lanes 5 & 6), yet is still unresponsive to Tat *trans*-activation (S1B and S1C Fig) and does not bind DNTTIP1 *in vitro* (Figs 1 and 2). The ATAC mutation responded in a manner indistinguishable to that of the ATCC mutation, further confirming that loss of DNTTIP1 was responsible for the phenotypes observed (Fig 7C). Because the DNTTIP1 binding site overlaps the binding site for bHLH transcription factors including AP4 and USF1 (Fig 1A), we employed mutations that

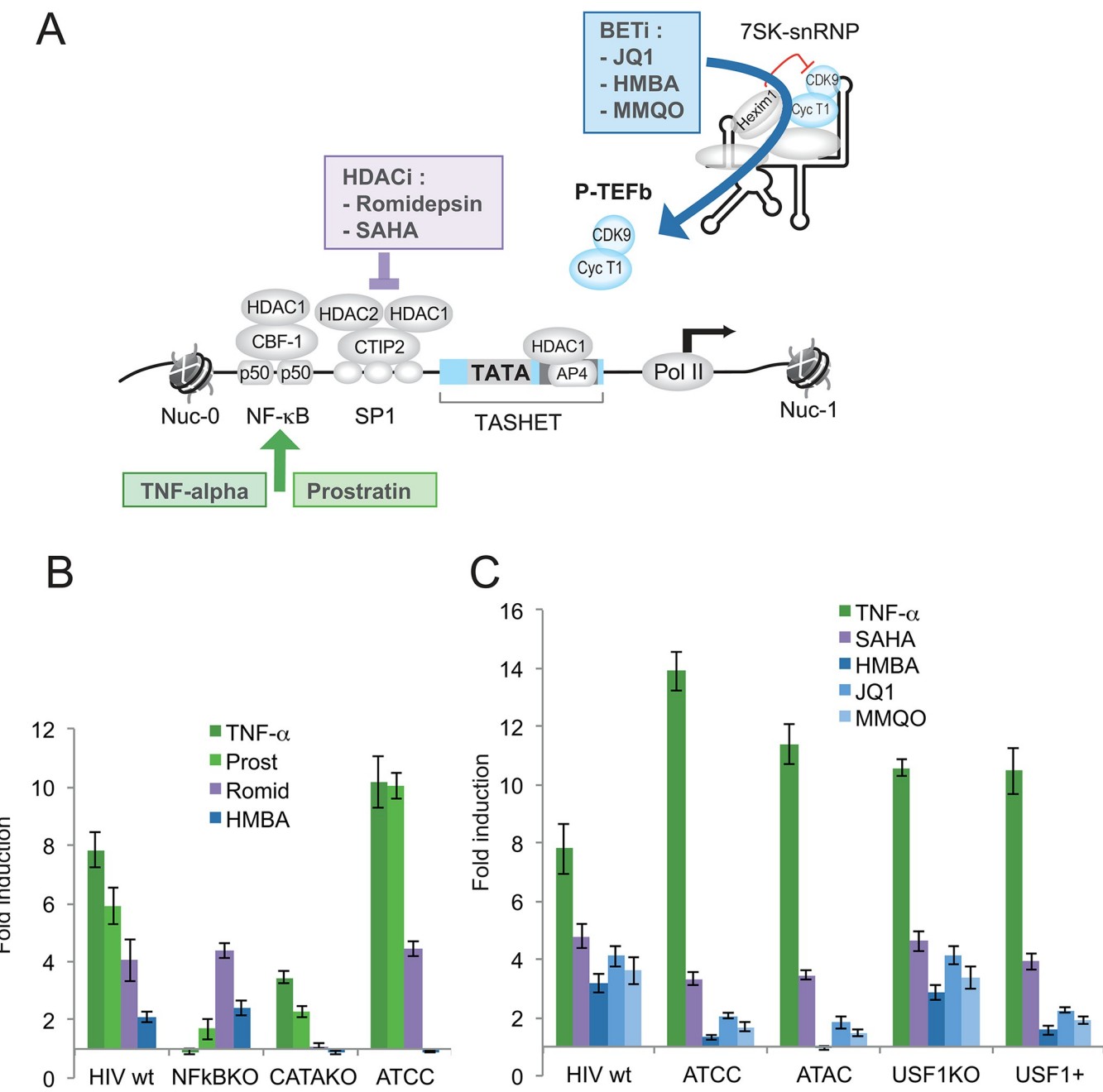

**Fig 7. Latency reversing agents (LRA) acting through the P-TEFb/7SK axis require the DNTTIP1 binding sites of TASHET.** A) A schematic representation of the major LRA pathways acting through NF-κB (green), HDACs (purple) or the P-TEFb/7SK axis (blue). B) Impact of LRAs on reporter viruses. Jurkat cells were infected with single-round reporter viruses bearing mutations in the HIV LTR (indicated on the x-axis), and then treated with LRAs for 16 hours before harvesting cell extracts for luciferase assays. The activity of integrated proviral HIV-1 promoter was monitored by luciferase activity. The results, calculated as RLU/μg of total protein are expressed as fold induction over the DMSO vehicle control for a given virus (y-axis). LRA color codes correspond to those in panel A. C) As in B, with additional LRA treatments and LTR mutations. Error bars represent the standard deviation of three independent experiments performed in parallel.

we have previously characterized [25] to separate the binding of bHLH proteins from that of DNTTIP1. Mutations that block binding of USF1 (USF KO) responded to LRAs essentially as the wild type promoter (Fig 7C), showing that USF1 binding is dispensable for LRA responsiveness. Conversely, mutations that enhance binding of USF1 (USF1+) [25], responded more

poorly to HMBA, JQ1 and MMQO (Fig 7C), reinforcing the conclusion that DNTTIP1 but not USF1 binding is essential for activation by these LRAs. Taken together, our findings demonstrate that intact DNTTIP1 binding sites within the HIV core promoter are required for the activation of HIV gene expression in T cells by small molecules that act via the P-TEFb/7SK axis.

## Discussion

The TATA box and adjacent sequences of HIV essential for Tat *trans*-activation (TASHET) plays a crucial and selective role in HIV gene expression [12], but the underlying molecular mechanisms have remained elusive. Our results reveal that TASHET is specifically recognized by the host cell mitotic histone deacetylase complex (MiDAC). MiDAC was identified in a proteomic screen designed to identify cellular targets of HDAC inhibitors [35]. MiDAC has been shown to contain protein subunits MIDEAS, HDAC1, HDAC2 and DNTTIP1 in myeloid leukemia K562 cells and human T lymphocyte CEM cells [36]. Recent work has shown that MiDAC is essential for embryonic development in mice and forms a unique multivalent structure [70]. Furthermore, MiDAC has been implicated in gene expression during neurite outgrowth in mice [71] and the negative regulation of the H4K20ac histone modification [72]. Our findings that 1) DNTTIP1 directly recognizes TASHET DNA *in vitro* and *in cellulo* (Figs 3 and 4), 2) DNTTIP1, MIDEAS, and NAT10 are required for full HIV transcription (Figs 5 and 6), and 3) NAT10 associates with MIDEAS (Fig 2), together with published data showing that NAT10 interacts with HIV Tat [38,39], support a model in which MiDAC is recruited to TASHET via the binding of its DNTTIP1 subunit (Fig 8). Our data are also consistent with a model that includes the recruitment of NAT10 to TASHET, particularly in the G2/M phase of the cell cycle, via a network of interactions with MIDEAS, where it can then bind directly to HIV Tat to activate transcription (Fig 8). The interaction of Tat with NAT10 can also help

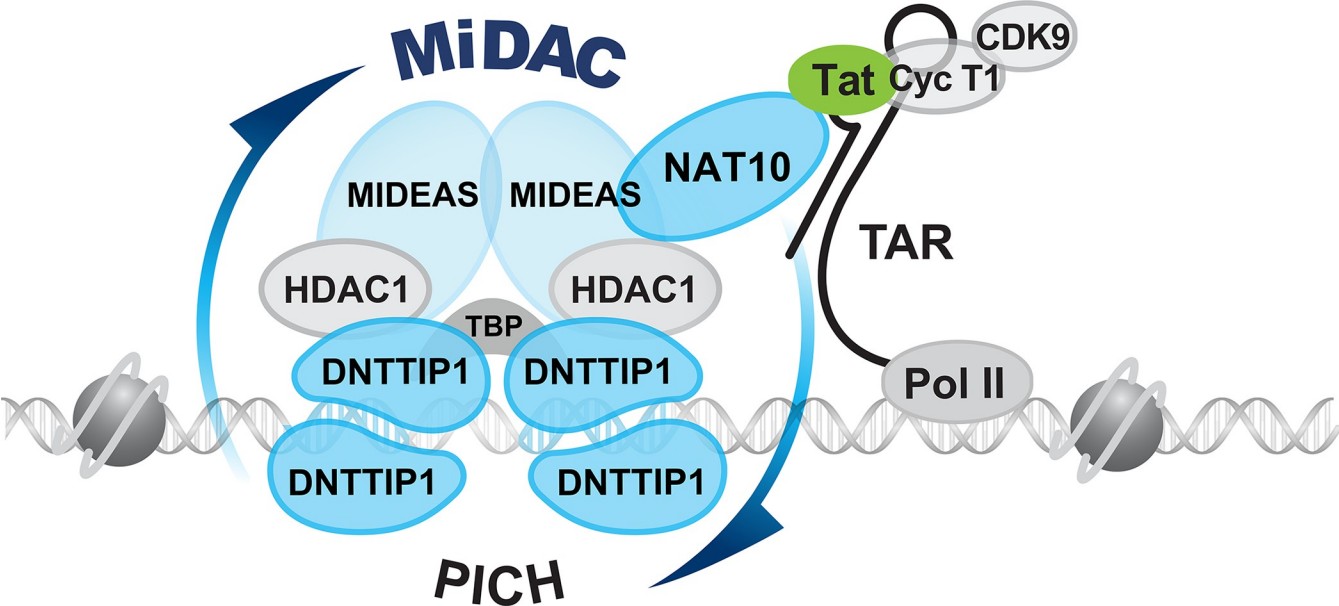

**Fig 8. Working model.** A hypothetical model depicting the binding of the mitotic deacetylase complex (MiDAC) to TASHET of the HIV core promoter. Host cell regulators DNTTIP1, MIDEAS and NAT10 identified in this work are in blue. Multimeric DNTTIP1 recognizes the CTGC motifs flanking the TATA box, likely via its SKI/SNO/DAC domain containing a HTH-like motif, and the TATA box via its AT-hook domain. DNTTIP1 interacts strongly with MIDEAS to recruit HDAC1/2. NAT10 is also recruited via MIDEAS and directly contacts HIV Tat to enhance HIV transcription.

explain the observation that Tat can be recruited to the HIV LTR in the absence of TAR RNA *in cellulo* [73] and, most importantly, provides the first mechanistic explanation for the specific requirement for TASHET in Tat-activated transcription.

DNTTIP1 (TdIF1) was originally identified in a two-hybrid screen as a nuclear protein that interacts with terminal deoxynucleotidyltransferase (TdT), a lymphocyte-specific enzyme involved in enhancing Immunoglobulin (Ig) and T cell receptor (TCR) diversity [74]. Building on their identification of DNTTIP1, Koiwai and coworkers reported that the AT-hook is involved in the recognition of AT-rich DNA, and that a predicted helix-turn-helix (HTH) is important for the recognition (Fig 3A) of the 5'-G**NTGC**ATG-3' consensus sequence defined by SELEX [47,50]. By analogy with these findings, we propose a model in which the AT-hook of DNTTIP1 binds primarily to the HIV TATA box while the HTH motif could interact with the CTGC motifs (Fig 3A). A role for DNTTIP1 in transcription was shown by placing DNTTIP1 consensus sites 5' of a minimal promoter where they produced increased reporter gene expression in response to DNTTIP1 overexpression [50]. ChIP-Seq was also employed to define genomic DNTTIP1 binding sites [51], including sequences that align with HIV TASHET (Fig 3A). A clear biological role for DNTTIP1 has not yet been defined, but gene ontology analysis has hinted at a potential role in ossification [51]. DNTTIP1 is encoded by a non-essential gene at the cellular level [75] and has been shown to positively regulate its own expression [51]. Structural studies have shown that DNTTIP1 contains an amino-terminal dimerization domain with an unusual structural fold that interacts directly with HDAC1 and a more carboxy-terminal domain structurally related to the SKI/SNO/DAC domain [42]. The data herein reveal that DNTTIP1 directly and selectively binds to TASHET, and that DNTTIP1 plays a central role in the recruitment of MiDAC and NAT10 to the HIV core promoter (Fig 8).

MIDEAS, like DNTTIP1, is a non-essential gene at the cellular level [75] and interacts with DNTTIP1 via its ELM/SANT domain [42]. Apart from the fact that SANT domains have been broadly implicated in chromatin remodelling, little is known about MIDEAS. Our data show positive roles for DNTTIP1, MIDEAS and NAT10 in basal and Tat-activated HIV transcription. Although HDACs were historically labeled as negative regulators of transcription, more recent work has revealed highly dynamic acetylation/deacetylation of histones during the transcription cycle [76], a positive role for HDACs in transcription elongation [77], and HDAC association with active promoter regions [78]. Our results demonstrate a net positive role for MiDAC and NAT10 in HIV transcription and pave the way to exploiting these findings to develop new therapeutic strategies to manipulate HIV latency via MiDAC components. Indeed, recent progress in the structural characterization of MiDAC complexes has revealed a SKI/SNO/DAC DNA binding domain (DBD) of DNTTIP1 that contains the HTH motif (Fig 3A), and that MiDAC HDAC activity is directly regulated by the binding inositol-1,4,5,6-tetra-kisphosphate (IP4) to the MIDEAS-HDAC1 interface [42]. Such detailed structural information is relevant for the development of compounds that selectively target MiDAC [79]. The data presented here raise the possibility that the specific targeting of MiDAC could generate LRAs with improved specificity over the broad spectrum HDAC inhibitors that have failed to reduce latent HIV reservoirs in clinical trials [80].

Our data show that NAT10 is recruited to TASHET *in vitro* with MiDAC (Fig 2), particularly from G2/M-arrested cell extracts. NAT10 was initially identified as a transcriptional regulator of hTERT in a yeast one-hybrid screen [37]. More recently, growing evidence supports the conclusion that the major catalytic activity of NAT10 is the acetylation of cytidine in RNA to form N4-acetylcytidine (ac4C) [81,82]. Biochemical [83] and structural [84] research showed that tRNA (Met) cytidine acetyltransferase (TmcA), a bacterial homolog of NAT10 is an ATP-dependent RNA helicase and acetyltransferase that acetylates tRNA. Furthermore,

NAT10 (RRA1 or KRE33 in yeast) has been shown to acetylate 18S rRNA in *Saccharomyces cerevisiae* [85,86], *Schizosaccharomyces pombe* [87], and humans [86,88]. mRNAs have also been shown to be acetylated by NAT10 in human cells [89] (reviewed in [90]). Significantly, ac4C mass spectrometry data supports the possibility that the RNAs of the positive sense viruses Zika virus, Dengue virus, hepatitis C virus (HCV), poliovirus and HIV-1 contain the ac4C modification [91]. Moreover, in the case of enterovirus 71, the proposed NAT10-dependent N4-acetylcytidine of viral transcripts enhances translation and stability of viral RNA. [92]. Intriguingly, recent work suggests that NAT10 can modify HIV RNA and play a positive role in HIV gene expression at least in part by stabilizing HIV RNA [93]. Functionally, NAT10 depletion has been reported to result in the accumulation of cells in G2/M [94], defects in the processing of 18S rRNA [86, 88], and impaired RNA-mediated transcriptional activation [95]. Most importantly for activated HIV gene expression, NAT10 has been shown to bind to HIV Tat in two independent and unbiased proteomic screens [38, 39]. Like HIV Tat [30], NAT10 is localized principally in the nucleolus [94]. Our finding is the first to demonstrate that NAT10 can associate with the HIV core promoter in the absence of Tat suggests a model in which NAT10 can be recruited to the HIV promoter before the production of viral RNA or Tat. Our work will support efforts to elucidate the mechanistic role of NAT10 in Tat *trans*-activation and HIV gene expression and more globally the role of ac4C in viral pathogenesis and innate immune sensing of viruses.

In a model of latency, mutation of the CTGC motifs required for MiDAC and NAT10 recruitment to the HIV core promoter eliminated its reactivation by LRAs that act through the P-TEFb/7SK axis (Fig 7). In contrast, the mutation of CTGC sequences only modestly reduced the response to LRAs of the HDAC inhibitor class (eg. SAHA, Fig 7). The fact that the HIV LTR can be activated by HDAC inhibitors in the absence of MiDAC recruitment to TASHET likely results from class I HDAC recruitment via multiple transcription factors including AP4, YY1, CBF-1 [96] and CTIP2 [97]. Our previous [25] and current findings argue against a major role for the binding sites for transcription factors USF1 and AP4 within TASHET in Tat-activation and latency reactivation (Fig 7). The selective impact of HIV LTR mutations that impair MiDAC recruitment to TASHET on reactivation by LRAs that act through the P-TEFb/7SK implies a key role for these complexes in the control of activated HIV gene expression.

The DNA sequence immediately downstream of the HIV TATA box contains densely overlapping binding sites for numerous cellular transcription factors. In addition to DNTTIP1, USF1, AP4, and conceivably other bHLH family members [98], the ZASC1 transcription factor has been shown to bind 3' of the HIV TATA box and to contribute to TAR independent Tat/P-TEFb recruitment [99]. DNTTIP1 and ZASC1 bind to overlapping sequences. The development of point mutations that separate the binding of ZASC1 from that of DNTTIP1 will be necessary to define the relative contributions to of ZASC1 and DNTTIP1 to HIV transcription in the future.

A unifying theme emerging from our findings is the link between TASHET-binding proteins and cell cycle control. MiDAC activity increases in the G2/M phase of the cell cycle [35] and NAT10 depletion result in accumulation of cells in G2/M [94]. Moreover, DNTTIP1 and MIDEAS have been found in complexes with Cyclin A, and both DNTTIP1 and MIDEAS undergo Cdk-dependent phosphorylation [100]. Genetically, unbiased functional genomic screens further show that DNTTIP1 and MIDEAS can both impact cell cycle progression [101,102]. Our results are compatible with previous findings that HIV transcription has peaks in the G2/M phase *in cellulo* [103] and *in vitro* [104]. The HIV accessory protein Vpr causes a blockage of cells in G2/M [105] and consequently activates HIV transcription, in part via the HIV TATA region [106,107]. The mechanism responsible for core promoter dependent

activation of HIV by Vpr remains unknown. Given our results, MiDAC emerges as a potential candidate for linking Vpr to LTR activity, and experiments to investigate such a roll are underway in our laboratory. The emerging links between cell cycle control and HIV gene expression that our data reveal are of practical importance for cure strategies that depend on fully activating or silencing HIV gene expression. As a concrete example, since one problem with the shock and kill approach has been that no more than 5% are actually reactivated by latency reversing agents [108], it may be necessary to combine these strategies with pharmacological manipulation of the cell cycle to improve clinical outcomes.

In addition to Vpr, numerous viral and cellular signals can modulate HIV transcription in a core promoter dependent manner (summarized in S2 Table). Moreover, multiple proteomic, transcriptomic and clinical data evoke potential links between HIV biology and the functions of DNTTIP1, MIDEAS and NAT10 (summarized for brevity in S3 Table). The discovery of a central role for these proteins in HIV transcription opens the door to experiments designed to exclude or confirm the clinical relevance of these potential links.

In conclusion, the data presented here uncover a functional role for the host cell mitotic histone deacetylase complex (MiDAC) components DNTTIP1, MIDEAS, as well as the Tat-interacting protein NAT10, in HIV transcription and latency. Our results bring to light the selective recognition of TASHET by DNTTIP1 to recruit MIDEAS, HDACs 1 & 2 and NAT10. NAT10 can then physically bridge the promoter to HIV Tat (Fig 8). The impact of these proteins on HIV transcriptional activation identifies them as potential drug targets to therapeutically control HIV latency in the quest for a cure or remission for HIV.

## Material & methods

### Ethics statement

The institutional review board (Comité d'éthique de la recherche du CIUSSS de l'Estrie) at the authors' institutions approved the study. All participants who donated blood for PBMC isolation provided written informed consent for their participation in the study (only adult participants were enrolled).

### Antibodies

Antibodies used in this study were raised against: DNTTIP1 (A304-048A, IP/WB). C14ORF43 (= MIDEAS; A303-157A-1, IP/WB) from Bethyl (Montgomery, TX); HDAC-1 (Ab7028), HIV1-Tat (Ab43014), from Abcam (Toronto, ON); AP-4 (sc-377042), DNTTIP1(sc-166296x for WB), USF-1 (sc-229), all from Santa Cruz, CA; Pol II (8WG16) from Covance (Emeryville, CA); Flag tag (F3165) and C14ORF43 (HPA003111 for WB) from Sigma-Aldrich. Monoclonal antibodies against TFII-D subunits were generous gifts from Dr. Lazlo Tora and have been described: TBP (3G3) [109].

Normal rabbit IgG (#2729, Cell Signaling, Danvers, MS) was used as specificity control in IP.

### Plasmids

Several plasmid constructs have been previously described: pCMV-Tat [110] and its empty control pCMV-ΔTat [25], wild type and LTR-mutated pNL4.3-LucE- [25]: the pNL4.3-Luc-ATCC mutation corresponds to CTGC5'3' mutation described in Wilhelm et al. [25]; the pNL4.3-Luc-mκB (NFκBKO) has been described [111]. The pNL4.3-Luc-ATAC has been cloned following the same strategy [25]. pLTR-YFP was obtained by subcloning the XhoI-NcoI portion of pNL4.3-LucE- in phRLnull. The Rluc sequence was then replaced by the YFP

sequence of pEYFP-N1 (Clontech, Mountainview, CA) by insertion into the NcoI-NotI sites. pLenti-DNTTIP1-Flag was constructed by PCR-based addition of a BamHI site in 5' and in frame Flag sequence followed by EcoRI site in 3' of the DNTTIP1 cDNA sequence. The resulting PCR product was then inserted in the same sites of pLenti6V5adapt (kindly provided by Dr. Nathalie Rivard, Université de Sherbrooke, Qc, Canada). pET28a-His-DNTTIP1 has been generated by in frame insertion of DNTTIP1 cDNA into BamH1/Xho1 sites of pET28a(+) (Novagen, kindly provided by Dr. Martin Bisaillon, Université de Sherbrooke, Qc, Canada) downstream of 6xHis tag sequence.

## Cell culture, nuclear extracts and immunoprecipitation

HeLa, HEK-293 and PBMC culture conditions were as previously described [25]. Jurkat E6-1 (ATCC #TIB-152), MOLT-4 (ATCC #CRL-1582) and THP-1 (ATCC #TIB-202) were grown in RPMI 1640 supplemented with 10% FBS. Nuclear Extracts were prepared from HeLa cells [112] or lymphoid cells [113] as previously described [25]. IP was performed with Surebeads protein G magnetic beads (Biorad, Hercules, CA). Briefly, 0.75 to 1.5μg of antibody was immobilized by incubation with 15μl of beads in IP100 buffer [114] without NP40. After extensive washing, 125μg NE were added overnight at 4°C with gentle rocking in IP100 buffer containing 0.05% NP40. After extensive washes with IP150 buffer, complexes were analyzed by SDS-PAGE and immunoblotting.

## DNA-affinity chromatography for protein identification by mass spectrometry

The affinity enrichment of factors bound to wt or mutated TASHET DNA sequence was performed using Dynabeads M-280 Streptavidin magnetic beads. 50μl Dynabeads were washed in 100μl B&W buffer according to the supplier's recommendations and the interaction with 50pmol biotinylated double-stranded oligonucleotide was performed for 15 minutes in 200μl B&W buffer (sequences in S1 File). After 3 washes in the same buffer, the DNA-coated beads were added to the interaction mixture (BB 1x: 20mM HEPES pH7.9, 100mM KCl, 5mM, 10% Glycerol; supplemented with 160μg acetylated-BSA (Promega R396A), 1.44nmol ds oligoB, 67.2μg ds poly(dIdC)-poly(dIdC) (Sigma P4929), 1mg HeLa nuclear extract (NE) and 0.05% CHAPS in a final volume of 500μl. After a 5 minutes interaction at room temperature, beads were washed in BB 1x, followed by three washes in BB 1x containing 250nM KCl and 0.02% NP-40. Beads were resuspended in Laemmli buffer, boiled, and the eluted denatured proteins were separated by SDS-PAGE.

## Silver staining

Silver staining was performed to visualize the proteins enriched by DNA-affinity chromatography. The gel was first dehydrated in 50% methanol, rinsed in water and stained for 15min (0.075% NaOH, 1.4% $NH_4OH$, 0.8% $AgNO_3$), washed again and revealed by a solution containing 0.005% citric acid and 0.02% formaldehyde. After rinsing, the revelation was stopped by a 7% acetic acid solution.

## Sample preparation for mass spectrometry analysis

The whole procedure was performed with methanol cleaned material in a dust free atmosphere. After resolution by SDS-PAGE, the TASHET DNA affinity enriched proteins were revealed by negative staining [115]. The bands of interest were cut out of the gels from purification on the wild-type promoter where bands were specifically visible and $ZnSO_4$ was

removed by two 8 min washes in 1ml of 25mM Tris, 192mM Glycine pH8.3, 30% acetonitrile (ACN). Further washing was performed by two 10 min incubations in 50mM $NH_4HCO_3$. Gels spots were dehydrated twice in 200μl ACN, followed by evaporation. 15μl sequencing grade modified trypsin (10μg/ml in 25mM $NH_4HCO_3$) were added and digestion was completed for 16-24h at 37˚C. Peptides were extracted by incubation in 50μl 50% ACN / 5% trifluoroacetic acid twice for 1h. Combined extracts were evaporated in a speed vac down to 5–10μl.

### Protein identification by LC-MS/MS and data analysis

The digested proteins from each gel spot were analyzed using the NanoLC Ultra 2Dplus System (SCIEX) combined with the cHiPLC system in Trap-Elute mode. The peptides were loaded (5μL injection volume) onto the trap (200 μm x 500 μm ChromXP C18-CL, 3μm, 300 Å) and washed for 10 min at 2μL/min. Then, a gradient of 8–35% acetonitrile (0.1% formic acid) over 15 min was used to elute the peptides from the trap on to the analytical column (75μm x 15cm ChromXP C18-CL, 3μm, 300 Å). Total run time was 45 min per sample. The column eluent was directly connected to the NanoSpray III Source for ionization on the TripleTOF 5600 System (SCIEX). Data were collected in data dependent mode using a TOF MS survey scan of 250 msec followed by 20 MS/MS with an accumulation time of 50 msec each. Data files were processed using ProteinPilot Software 4.0 (SCIEX) with integrated false discovery rate (FDR) analysis [116]. Further data analysis was performed using the ProteinPilot Protein Alignment Template (SCIEX) (https://sciex.com/software-support/software-downloads).

### Recombinant protein production and purification for circular dichroism analysis

Expression of recombinant N-terminally 6xHis tagged DNTTIP1 was induced in BL21 DE3 E. coli strain by 0.5mM IPTG at 16˚C for 12h. All following steps were performed at 4˚C or on ice. Crude extract was obtained from a 500ml culture bacterial pellet by successive sonications in 2x20ml buffer A (30mM Tris pH8.0, 10% Glycerol, 1M NaCl, 10mM Imidazole pH8.0, 1x Protease Inhibitors Cocktail (Roche Complete Mini EDTA free), 0.4mM PMSF, 7mM β-mercaptoethanol). His-tagged protein was purified on a 1ml Ni-NTA column (Qiagen) prewashed in buffer A containing 500mM imidazole w/o β-mercaptoethanol. The column was washed with buffer A containing 60mM imidazole, and elution was performed in fractions with buffer A containing 250mM imidazole and 50mM of each L-Arg and L-Glu amino acids for protein stabilisation. Fractions containing pure protein were pooled and dialysed against the appropriate buffer for subsequent experiments (see S1 File).

### Circular Dichroism (CD)

CD spectra and denaturation curves were recorded on a Jasco J-810 spectropolarimeter equipped with a Jasco Peltier-type thermostat. 25μM purified 6xHis-DNTTIP1 was loaded into a 1mm path length quartz cuvette with or without 25μM doubled-stranded oligonucleotides in CD buffer (10nM Na.PO4 pH6.5, 100mM KCl, 1mM TCEP). Spectra were recorded by accumulating 10 scans at 20˚C from 250 to 210 nm with a 1nm band width beam at 0.2nm intervals. Thermal denaturation curves were obtained at 222nm from 20 to 80˚C at a 1˚C/min rate with 0.2˚C intervals. The DNA contribution to the CD signal was subtracted from the denaturation curves containing DNA (sequences in S1 File).

### Fluorescence anisotropy

Fluorescence anisotropy measurements were performed on a F-2500 Fluorescence

Spectrophotometer (Hitachi) equipped with polarizers. 25-mer double stranded DNAs labeled with fluorescein (sequences in S1 File) were diluted at 15nM in anisotropy buffer (10mM Na.PO4 pH6.5, 150mM KCl, 250μM TCEP, 5% glycerol) and anisotropy was measured with gradual increase in recombinant protein concentration. Each protein addition was followed by a 5 min incubation in order to reach equilibrium. Anisotropy values (r) were calculated as described [117].

## Virus production

pNL4.3 based viral particles have been produced as described [25]. Lentiviruses for DNTTIP1-Flag expression were produced following the same protocol. Briefly, HEK-293 cells were transfected with 7μg plenti-DNTTIP1-Flag (or the empty plenti6V5adapt plasmid), 2.5μg pLP1, 2.5μg pLP2, and 2μg pLP/VSVG per one 100mm dish. Virus particles were harvested 48 to 72h later, treated with 10U DNase-1 (Sigma DN25) for each μg of transfected plasmid for 30min à 37°C in presence of 10mM $MgCl_2$ and stored at -80°C.

TRC2 (Sigma) lentiviruses (for target sequences see S1 File) for shRNA delivery were produced by transfecting HEK-293 cells in a 60mm dish with a 1ml OptiMEM (Gibco) mix containing 15μl of Lipofectamine-2000 (Invitrogen) and 1.5μg of each plasmid (pLP1, pLP2, pLP/VSVG, TRC2 construct). The medium was replaced by complete medium 4h post-transfection and virions were collected as above.

## siRNA depletion and HIV reporter assays in HeLa cells

HeLa cells were transfected 12h after seeding with 20nM dsiRNA (for target sequences see S1 File) with lipofectamine 2000 according to the manufacturer recommendations. 36h later, the reporter system was transfected as previously described [25]. Briefly, 250ng of pLTR-YFP reporter and 50ng of pCMV-Tat or pCMV-ΔTat were transfected with lipofectamine 2000. 24h later, cells were collected for flow cytometry and Western blot analysis.

## shRNA depletion and HIV reporter assays in MOLT4 cells

$10^6$ MOLT4 cells were infected with 40μl of TRC lentivirus and maintained at $10^6$ cells/ml overnight. The next day, the medium was replaced and cells were diluted at $5.10^5$/ml. 48h after infection, puromycin selection was applied (0.75μg/ml). 4 days later, resistant cells had recovered from the selection and were infected with pNL4.3 reporter virus as described [25] and collected 48h later for luciferase assays as described below.

## Stable cell line production

HeLa cells were infected as previously described [25] with pLenti-DNTTIP1-Flag or its control. 3 days post infection, 5μg/ml of blasticidine was added. The cultures were maintained under selection for 2 to 3 weeks, until toxicity was no longer observed.

## Chromatin immunoprecipitation (ChIP)

Cells were infected with pNL4.3 virus at a MOI of 0.8 for 48h. HeLa cells were crosslinked by direct addition of 1% PFA to the culture medium for 10 min at room temperature. 125mM Glycine was added to stop the crosslink. Cells were washed twice with cold PBS, harvested on ice, and frozen into aliquots of $8.10^6$ cells. Chromatin IPs were performed as previously described [118] except that soluble chromatin was obtained by MNase digestion [119] with the following changes: cell wash buffer I contained 0.5% Triton X100; $8x10^6$ crosslinked HeLa cells were digested by 80U MNase, yielding 50–70μg soluble chromatin for 1 IP. 4μg of the

appropriate antibody and 20µl of Magnetic protein A/G beads (Magnachip, EMD-Millipore, Billerica, MA, USA) were used for the IP step, according to the manufacturer's instructions.

## LRA treatment and luciferase reporter assay

Jurkat cells were infected with pNL4.3 based viruses corresponding to an estimated MOI of 1 for 36h before the addition of LRA (detailed in S1 File) for another 16h. Cells were collected and lysed in Passive Lysis Buffer (Promega, Madison, WI, USA). The luciferase activity was measured on a Lumistar Galaxy (BMG Labtechnologies, Ortenberg, Germany) after the addition of an equal volume of 2x luciferase buffer (20mM Tris-HCl pH7.8, 1.07mM MgCl2, 2.7mM MgSO4, 0.1mM EDTA, 33.3mM DTT, 470µM D-Luciferin (E1601, Promega, Madison, WI, USA), 270µM Co-enzyme A (C3019, Sigma), 530µM ATP (A7699, Sigma)). The number of proviruses per cell was assessed by viral-specific qPCR as previously described [25].

## Supporting information

**S1 Data. Raw numerical data.** Excel spreadsheet containing, in separate sheets, the underlying numerical data and statistical analysis for Figs 3C, 3D, 3E, 4C, 5B,5C, 5D, 6B, 7B, and 7C. (XLSX)

**S1 Fig. EMSA and transcriptional activity profiles of the ATCC vs ATAC mutation.** A) EMSA comparing the PIC profile obtained with different probes (lanes 1–2: wt; 3–4: ATCC mutation; 5–6: ATAC mutation) with Jurkat (odd lane numbers) and HeLa (even lane numbers) cell line nuclear extracts (NE). aPIC = aberrant PIC. The stars indicate complexes obtained only with Jurkat NE. B) HIV promoter activity measured in HeLa cells after transfection of reporter plasmids bearing the same mutations as in A, with (darker grey) and without (lighter grey) Tat coexpression. Tat *trans*-activation is the ratio between +Tat and -Tat normalized to wt (100%). C) HIV promoter activity measured in Jurkat cells after infection with reporter pseudotyped viruses bearing the same mutations. (EPS)

**S2 Fig. Selected MS/MS spectra supporting the protein identifications from gel spots.** LC-MS/MS analysis was performed on digested proteins from gel spots and identified from acquired MS/MS spectra. Two MS/MS spectra from each protein are shown and annotated with the peptide fragmentation information for sp|Q9H147|DNTTIP1_HUMAN (A,B), sp|Q9H0A0|NAT10_HUMAN (C,D) and sp|Q6PJG2|MIDEAS_HUMAN (E,F). (PPTX)

**S1 Table. Peptide evidence for the proteins identified from gel spots.** Peptides with confidence >95% from the ProteinPilot software database search are listed for each of the identified proteins. Reported protein sequence coverage is based only on peptide with >95% confidence. Annotated MS/MS spectra for selected peptides is shown in S2 Fig. (XLSX)

**S2 Table. Modulators of HIV transcription that act in part via the TATA-box region.** (DOCX)

**S3 Table. Potential links of DNTTIP1, MIDEAS & NAT10 to HIV biology.** (DOCX)

**S1 File. Additional material and methods.** The file contains: 1) Oligonucleotides and target sequences; 2) DNTTIP1 purification dialysis buffers and protocol; 3) Latency Reactivating

Agents used in the study.
(DOCX)

## Acknowledgments

We thank Audrey Daigneault, Aurélie Delannoy, Catherine Desrosiers and Joannie St-Germain for technical assistance. We thank Nancy Dumais for gifts of Jurkat and MOLT4 cell lines, and Ms Anette Bald for isolating PBMCs from the peripheral blood of healthy donors. We are grateful to Albert Jordan for the generous gift of MMQO, viral vectors, and helpful discussions. We thank all the members of the Canadian HIV Cure Enterprise (CanCURE) consortium for helpful exchanges. We acknowledge the use of the Mitocheck database while conducting this study. We thank Annie Leclerc and Barbara Bell for assistance with artwork. We are grateful to Drs. Benoit Chabot, Craig McCormick and John Rhode for critical comments on the manuscript.

## Author Contributions

**Conceptualization:** Emmanuelle Wilhelm, Pierre Lavigne, Christie L. Hunter, Brendan Bell.

**Data curation:** Emmanuelle Wilhelm, Pierre Lavigne, Christie L. Hunter, Brendan Bell.

**Formal analysis:** Brendan Bell.

**Funding acquisition:** Brendan Bell.

**Investigation:** Emmanuelle Wilhelm, Mikaël Poirier, Morgane Da Rocha, Mikaël Bédard, Patrick P. McDonald, Christie L. Hunter, Brendan Bell.

**Methodology:** Emmanuelle Wilhelm, Mikaël Poirier, Morgane Da Rocha, Patrick P. McDonald, Pierre Lavigne, Christie L. Hunter, Brendan Bell.

**Project administration:** Brendan Bell.

**Resources:** Brendan Bell.

**Supervision:** Pierre Lavigne, Brendan Bell.

**Validation:** Mikaël Poirier, Brendan Bell.

**Visualization:** Brendan Bell.

**Writing – original draft:** Emmanuelle Wilhelm, Pierre Lavigne, Brendan Bell.

**Writing – review & editing:** Emmanuelle Wilhelm, Mikaël Poirier, Morgane Da Rocha, Pierre Lavigne, Christie L. Hunter, Brendan Bell.

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
