## [Decision Letter · Decision Letter 0]

30 Jan 2024

Dear Dr. Bell,

Thank you very much for submitting your manuscript "Mitotic deacetylase complex (MiDAC) recognizes the HIV-1 core promoter to control Tat-activated transcription and latency" for consideration at PLOS Pathogens. As with all papers reviewed by the journal, your manuscript was reviewed by members of the editorial board and by several independent reviewers. In light of the reviews (below this email), we would like to invite the resubmission of a significantly-revised version that takes into account the reviewers' comments.

First, let me apologize for the long time it took this paper to be reviewed.  Apparently, there were problems getting an editor assigned to it.  As you see below, the reviews of the paper are mixed.  Reviewers 1 and 2 are especially critical of the latency model you use and of the language used around it to describe the activity of the MiDAC complex in latency.  I agree with these comments that you would need to use more established true latency models to make these claims.  However, I also feel that if you refocus the description of these experiments as tests for HIV transcriptional activity rather than as true latency reversal (along with corresponding changes in the introduction and description), then they can still be presented and are still valuable results.

Reviewer #3 was especially interested in the role of mitosis on HIV LTR regulation given that you purified these extracts from mitotic cells, and feels that the manuscript would be greatly strengthened by an extension of experiments to examine the effects of the cell cycle on HIV transcription. 

The reviewers also made other comments to improve the paper.  I look forward to seeing a revision that addresses them.

We cannot make any decision about publication until we have seen the revised manuscript and your response to the reviewers' comments. Your revised manuscript is also likely to be sent to reviewers for further evaluation.

Sincerely,

Michael Emerman, Ph.D.

Guest Editor

PLOS Pathogens

Richard Koup

Section Editor

PLOS Pathogens

Michael Malim

Editor-in-Chief

PLOS Pathogens

orcid.org/0000-0002-7699-2064

Reviewer's Responses to Questions

**Part I - Summary**

Reviewer #1: This well-written manuscript from Wilhelm and colleagues investigates cellular proteins that can bind to the HIV-1 promoter (termed TASHET by Authors) and function to contribute to RNAP II and HIV-1 Tat transcription of the integrated viral genome. The proteins DNTTIP1, MIDEAS, and NAT10 were identified by virtue of their specific binding to the viral promoter and not mutant viral promoters. DNTTPIP1 and MIDEAS are subunits of a HDAC complex termed MiDAC whose activity increases during mitosis. NAT10, an acetyltransferase, has previously been reported to bind to the Tat protein. SiRNA depletions indicate that depletion of the three proteins individually can reduce expression from the viral LTR.

The Authors present clear evidence that DNTTIP1, MIDEAS, and NAT10 associate specifically with the viral promoter in vitro, and that DNTTIP1 does so in cells. They also present convincing evidence from their siRNA depletions that these factors can function to enhance HIV-1 LTR expression. However, the effects of siRNA depletions on a reporter plasmid are rather modest – generally, only a few fold at best. As described below, the “latency” reactivation experiment presented in Figure 6 is not up to the standard of the field. Also as described below, the impact of the study is limited without additional experiments from primary CD4+ T cells and patient cells harboring latent HIV-1.

Reviewer #2: Wilhem and colleagues use biotinylated oligonucleotides containing the WT TASHET sequence of HIV as determined by an earlier paper of the Bell group as well as oligos containing mutations within the GTGC motifs of TASHET to enrich for binding proteins from HeLa, Jurkat, and total PBMC extracts. Mass spec to identify proteins was specifically performed on material enriched from HeLa extracts and selective band excision based on visually identifiable differences was used rather than complex mixture, in solution analysis which may have resulted a more unbiased approach. They identified MIDEAS and DNTTIP1, and to a weaker extent NAT10, as novel binders of the TASHET sequence, members of the mitotic deacetylase complex (MiDAC).

The follow-up experiments to confirm binding to the oligos in question are well done, however the overall levels of enrichment observed by western blot are a bit underwhelming but appear specific. A stronger, validated binder such as TBP would been good to see as a positive control in the oligo enrichment experiments. The ChIP is well done with appropriate controls. We have concerns about appropriate controls in the later experiments and are unclear as to the proposed mechanism and when MiDAC functions in the HIV lifecycle as most experiments are undertaken with single round infections after 48hrs. We would like to see the points below addressed.

Reviewer #3: In this manuscript Wilhelm et al., identify and characterize a novel complex which binds the TATA box and adjacent sequences of the HIV-1 LTR essential for Tat trans-activation (TASHET). For this, they used sequence-specific DNA affinity chromatography where they purified a complex containing the factors DNTTIP1, MIDEAS, and NAT10. MIDEAS and DNTTIP represent components of the MiDAC histone deacetylase complex which is primarily active during mitosis. NAT10 is an acetyltransferase which was previously shown to interact with the HIV-1 viral transactivator Tat. Binding of this complex with the HIV-1 core promoter was shown to be dependent upon nucleotides near the TATA element required for responsiveness to Tat. Using HeLa cells, they demonstrated stronger overall binding of this complex with TASHET in cells arrested in mitosis, compared to an unsynchronized cell population. Using co-immunoprecipitation, they demonstrate that these proteins are associated in the absence of DNA, and that NAT10, at least also interacts with Cyclin T1. They used several biophysical techniques to determine that recombinant DNTTIP1 was capable of specifically binding TASHET DNA oligos in vitro. Using chromatin immunoprecipitation, they showed that Flag-tagged DNTTIP1 also bound the HIV-1 LTR in HeLa cells. Knocking down expression of these proteins in HeLa cells with siRNA inhibited HIV-1 expression in HeLa cells, both in the presence and absence of Tat. The largest effect is produced by DNTTIP1 and NAT10 siRNAs. They also examined the effect of shRNAs against DNTTIP1 on HIV-1 expression in the MOLT4 T cell line, where they observe ~50-60% reduced HIV-1 expression with three different shRNAs. Mutations that prevent binding of DNTTIP1 to the LTR, and inhibit activation by TAT, were also shown to inhibit reactivation by BET inhibitors, which are thought to work by encouraging release of P-TEFb from 7SK.

This is a very interesting report which employs some elegant biochemistry to identify a novel complex that interacts with sequences at the core HIV-1 LTR promoter required for activation by viral Tat. Association of this complex with this region of the LTR raises several important questions. First, many other proteins and complexes are known to bind this same region, not the least of which are the GTFs, RNA PolII, mediator, TFIID etc, but also numerous others, including YY1, LSF, AP4, ZASC1, and USF1/2. What seems to be unique about this complex is that its affinity towards the TASHET sequence is apparently stronger in cells arrested in mitosis. As such this manuscript would be strengthened considerably by a more detailed analysis of interaction with the LTR in vivo during the cell cycle. I don't believe anyone has examined effects of cell cycle on regulation of HIV-1 transcription, or TAT transactivation, and such an analysis would make this a very exciting study. Relating to this, strangely, I note that the authors don't really discuss the potential role of cell cycle for HIV-1 transcriptional regulation in the manuscript. Secondly, this putative complex contains both an HDAC and an acetyltransferase; the authors mention in the Discussion that acetylation/ deacetylation dynamics on histones may contribute to control of transcriptional elongation. For this manuscript it would be important to demonstrate the extent that the HDAC and acetylase activities are required for their effect on HIV-1 expression, and their effect on modification of histones at the LTR.

**Part II – Major Issues: Key Experiments Required for Acceptance**

Reviewer #1: 1. As described in Methods, the “latency reversal” experiment presented in Figure 7 involved an infection of Jurkat cells for 36 hours, followed by addition of LRAs. It is not possible to distinguish effects of the LRAs on ongoing viral replication vs latency, and latency is likely to be limited at 36 hours. There are Jurkat lines available with latent virus and reporter proteins that available for this type of experiment with mutant LTRs.

2. The impact of the study would be increased if the levels of DNTTIP1, MIDEAS, and NAT10 were examined in resting and activated primary CD4+ T cells. This may demonstrate some interesting regulation of these proteins.

3. The impact of the study would be increased with an in vitro analysis of the role of these proteins in reactivating latent virus from CD4 T cells from patients on suppressive ART. The analysis of patients’ cells is the standard for high impact studies of HIV-1 latency.

Reviewer #2: 1) Experiments validating the mass spec results are well done, but the evidence for NAT10 association is underwhelming. NAT10 was only confirmed in nocodozole treated cells and appears enriched in the ATCC IP as well, not only the HIV WT (Fig 2D). This would suggest NAT10 recruitment is not dependent on MIDEAS and DNTTIP1? Could the authors please address given later conclusions that NAT10 is a key lynchpin in why the ATCC sequence is not Tat-responsive.

2) There is current literature that DNTTIP1/MiDAC localization appears to correlate with levels of H3K27ac and H4K20ac at targeted promoters which control gene expression (https://pubmed.ncbi.nlm.nih.gov/32297854/). Given the authors have validated siRNAs targeting these proteins and have established ChIP protocols, we would highly recommend examining the HIV LTR for changes in these marks upon MiDAC component depletion. This would lend significant weight to a potential mechanism for positive regulation by MiDAC.

3) In Figure 5, the authors indicate the cells are transfected with an HIV LTR reporter plasmid and Tat expression plasmid after siRNA depletion. How was transfection efficiency controlled and standardized between samples to ensure all samples received the same amount of DNA? What do the error bars represent? Are these independent replicates performed at different times? This should be included in the figure legend and for all other figures containing error bars.

a. The meaning of the % Tat transactivation in this figure is extremely confusing. Does this indicate MFI relative to % positive YFP, or MFI of Tat+siRNA/NoTat+siRNA? Why is the % transactivation for NAT10 N2 80% but YFP+ cells decreased by more than 50%? Does this indicate higher MFI but in fewer cells, suggesting greater activation in a select number of cells?

4) Figure 6C – How was the number of integrated viruses per cell determined? This was not detailed in the methods. The methods as written suggest an infection at an MOI of 1 yet this figure indicates an average of 5 viruses/cell? How is this possible given the MOI?

5) Figure 7 – The authors cite work by the Verdin and Sadowski labs performed using dual-fluorescent reporters to justify that latency is established in 48hrs in single round infections in Jurkats. While the papers cited do show rapid silencing of a population, it is not the entire population and the specific latent population can be identified and sorted via the dual-reporter system to specifically examine the early latent pool. Using a luciferase-based virus, this population cannot be sorted and it is impossible to tell if the increased luciferase is due to increased transcription from those viruses that are not immediately silenced verses from those that are. Linking these sequences to reactivation from latency is a stretch.

a. The involvement of MiDAC in latency and reactivation to various LRAs would be better reinforced with DNTTIP1/MIDEAS knockdowns in established latency models (Verdin JLat lines, Karn cell lines), followed by treatment with various LRAs to examine viral reactivation in the absence of these proteins.

6) Overall, we are unclear as to the final mechanism being proposed. Is the primary function of MiDAC Tat-recruitment via scaffolding of NAT10? Does MiDAC induce chromatin changes at the LTR? Does MiDAC deacetylate protein targets? The ATCC mutated TASHET sequence shows significant differences in binding of proteins via EMSAs in Supplementary Figure 1 and in the Whilhem et al Retrovirology 2012 paper. While the ATAC sequence does appear more like wild-type with respect to protein binding, there does not appear to have been a truly unbiased approach to characterized all proteins bound to these oligos and there may be other changes to recruited complexes which impact overall Tat recruitment upon mutation of this core sequence. We agree that there does appear to be reduced Tat responsiveness when MiDAC components are specifically knocked down, but there is no definitive experiment that points to the mechanism of this action. Further, as everything is done in the context of a 48hr single round infection, this would suggest MiDAC is involved in aiding productive transcription in early viral infection but the relevance reactivation after long-

---

## [Editor Report · Decision Letter 1]

5 Apr 2024

Dear Dr. Bell,

We are pleased to inform you that your manuscript 'Mitotic deacetylase complex (MiDAC) recognizes the HIV-1 core promoter to control activated viral gene expression' has been provisionally accepted for publication in PLOS Pathogens.

Best regards,

Michael Emerman, Ph.D.

Guest Editor

PLOS Pathogens

Richard Koup

Section Editor

PLOS Pathogens

Michael Malim

Editor-in-Chief

PLOS Pathogens

orcid.org/0000-0002-7699-2064
---

## [Editor Report · Acceptance letter]

26 Apr 2024

Dear Dr. Bell,

We are delighted to inform you that your manuscript, "Mitotic deacetylase complex (MiDAC) recognizes the HIV-1 core promoter to control activated viral gene expression," has been formally accepted for publication in PLOS Pathogens.

Best regards,

Michael Malim

Editor-in-Chief

PLOS Pathogens

orcid.org/0000-0002-7699-2064